# Presynaptic GABA$_B$ receptors functionally uncouple somatostatin interneurons from the active hippocampal network

Sam A Booker[1,2†]*, Harumi Harada[3†], Claudio Elgueta[4], Julia Bank[3,5], Marlene Bartos[4], Akos Kulik[3,5]*, Imre Vida[1]*

[1]Institute for Integrative Neuroanatomy, Charité - Universitätsmedizin Berlin, Berlin, Germany; [2]Centre for Discovery Brain Sciences, University of Edinburgh, Edinburgh, United Kingdom; [3]Institute for Physiology II, Faculty of Medicine, University of Freiburg, Freiburg, Germany; [4]Institute for Physiology I, Faculty of Medicine, University of Freiburg, Freiburg, Germany; [5]BIOSS Centre for Biological Signalling Studies, University of Freiburg, Freiburg, Germany

***For correspondence:**
sbooker@exseed.ed.ac.uk (SAB);
akos.kulik@physiologie.uni-freiburg.de (AK);
imre.vida@charite.de (IV)

[†]These authors contributed equally to this work

**Abstract** Information processing in cortical neuronal networks relies on properly balanced excitatory and inhibitory neurotransmission. A ubiquitous motif for maintaining this balance is the somatostatin interneuron (SOM-IN) feedback microcircuit. Here, we investigated the modulation of this microcircuit by presynaptic GABA$_B$ receptors (GABA$_B$Rs) in the rodent hippocampus. Whole-cell recordings from SOM-INs revealed that both excitatory and inhibitory synaptic inputs are strongly inhibited by GABA$_B$Rs, while optogenetic activation of the interneurons shows that their inhibitory output is also strongly suppressed. Electron microscopic analysis of immunogold-labelled freeze-fracture replicas confirms that GABA$_B$Rs are highly expressed presynaptically at both input and output synapses of SOM-INs. Activation of GABA$_B$Rs selectively suppresses the recruitment of SOM-INs during gamma oscillations induced in vitro. Thus, axonal GABA$_B$Rs are positioned to efficiently control the input and output synapses of SOM-INs and can functionally uncouple them from local network with implications for rhythmogenesis and the balance of entorhinal versus intrahippocampal afferents.

## Introduction

The ability of cortical networks to process information requires a fine spatiotemporal balance of glutamatergic excitation and GABAergic inhibition. Inhibition in these circuits arises from GABAergic interneurons (INs), which target either the perisomatic or dendritic regions of principal cells (PCs) and are embedded in the local network in feedback or feedforward (*Booker and Vida, 2018*; *Klausberger and Somogyi, 2008*). A subpopulation of IN expresses the neuropeptide somatostatin (SOM) and innervates the distal apical dendritic tufts of PCs and INs (*Katona et al., 1999*; *McBain et al., 1994*). SOM-INs act as a ubiquitous inhibitory feedback element in hippocampal and neocortical circuits, due to their preferential excitatory input from local PCs (*Ali and Thomson, 1998*; *Blasco-Ibáñez and Freund, 1995*; *Lacaille et al., 1987*; *Müller and Remy, 2014*; *Shigemoto et al., 1996*; *Urban-Ciecko et al., 2018*; *Yuan et al., 2017*). Besides controlling dendritic excitability and synaptic plasticity at the single-cell level, hippocampal SOM-INs contribute to the co-ordination of population activity, particularly in the theta frequency range (4–12 Hz) (*Gloveli et al., 2005b*; *Klausberger et al., 2003*; *Maccaferri and McBain, 1996*), but also at higher frequencies in the beta and lower gamma band (*Chen et al., 2017*; *Hakim et al., 2018*). They gate information flow between and within cortical areas (*Leão et al., 2012*; *Naka et al., 2019*) and

support learning and memory processes (*Lovett-Barron et al., 2014*; *Abbas et al., 2018*; *Adler et al., 2019*).

Cortical INs are themselves strongly controlled by inhibitory mechanisms produced through both ionotropic GABA$_A$ receptors (GABA$_A$R) and metabotropic GABA$_B$Rs (*Kulik et al., 2018*). Indeed, SOM-INs in CA1 of the hippocampus have been shown to receive a strong GABA$_A$R-mediated synaptic input from local INs, dominantly those expressing calretinin (CR) and vasoactive intestinal peptide (VIP) (*Tyan et al., 2014*). GABA$_A$R-mediated inhibition onto INs more generally has been linked to disinhibitory network mechanisms (*Acsády et al., 1996*) and the emergence of coherent oscillatory network activity (*Bartos et al., 2002*; *Bartos et al., 2007*; *Traub et al., 1999*; *White et al., 2000*). In contrast, the contribution of GABA$_B$Rs to network functions is less well understood (*Booker et al., 2013*; *Brown et al., 2007*; *Kohl and Paulsen, 2010*), despite the ubiquitous expression and abundant distribution of GABA$_B$Rs at both pre- and postsynaptic locations on PCs and INs alike (*Kulik et al., 2006*; *Kulik et al., 2003*). We have recently shown that SOM-INs in CA1 express high levels of postsynaptic GABA$_B$Rs and their activation does not activate Kir3 channels, but inhibit dendritic L-type (Ca$_v$1.2) calcium channels to suppress synaptic plasticity (*Booker et al., 2018*). This result raises the question: do presynaptic GABA$_B$Rs contribute to short-term, direct inhibition of synaptic transmission at the input and output synapses of SOM-INs, leading to circuit level disinhibition? In fact, presynaptic GABA$_B$R-mediated inhibition would be able to efficiently control IN recruitment and to functionally uncouple them from the local network (*Huh et al., 2016*; *Urban-Ciecko et al., 2015*).

In this study, we first characterized the surface expression, localization and function of presynaptic GABA$_B$Rs at input and output synapses of SOM-INs in CA1 of the rodent hippocampus by combining in vitro electrophysiology with optogenetics and quantitative immunoelectron microscopy. We then assessed the impact of GABA$_B$R activation on network recruitment of SOM-INs during pharmacologically induced network oscillations in vitro.

## Results

### Presynaptic GABA$_B$Rs Strongly Inhibit Glutamatergic and GABAergic Synaptic Inputs onto SOM-INs

As SOM-INs largely lack postsynaptic GABA$_B$R mediated K$^+$-currents (*Booker et al., 2018*), we hypothesized that this receptor may confer presynaptic inhibition at synaptic inputs onto SOM-INs from CA1 PCs. Therefore, we performed whole-cell recordings from the INs combined with extracellular stimulation of their inputs in the *alveus* in rat acute hippocampal slices (*Figure 1*). We identified SOM-INs during the recordings as neurons having (1) their soma located at the *str. oriens/alveus* border, (2) high-frequency, but accommodating discharge pattern and (3) a large 'sag' in response to hyperpolarizing current steps during the recordings (*Figure 1A*). After the recordings, immunoreactivity for SOM was confirmed in 63 visualized INs (*Figure 1A*; *Booker et al., 2018*); cells that were negative for SOM were excluded. Most visualized SOM-INs in rat slices showed characteristic horizontally-oriented dendrites restricted to the *str. oriens/alveus* and 25 cells (39.7%) possessed an axon projecting to the *str. lacunosum-moleculare* (*Figure 1A*), consistent with the morphological features of oriens/alveus-lacunosum-moleculare (OLM) INs as described previously (*McBain et al., 1994*; *Katona et al., 1999*). Further 2 INs (3.2%) had bistratified axons in *str. radiatum* and *oriens*, 4 (6.4%) had axons confined to *str. oriens* only. The remaining INs either had axons cut close to the soma (8 cells, 12.7%) or were not sufficiently filled to allow morphological identification (24 cells, 38.1%). This division of cell identity is comparable to that we have observed previously (*Booker and Vida, 2018*). In an additional set of recordings from mouse slices (see below), we identified 15 SOM-INs, of which 7 were OLM cell (46.7%), 2 were bistratified cell (13.3%), the remainder had a cut axon (1 cell) or not filled sufficiently (5 cells) to be identifed morphologically.

In the presence of GABA$_A$R antagonists (gabazine or bicuculline, both 10 µM), *alveus* stimulation produced short-latency EPSCs with an average amplitude of 77.7 ± 16.1 pA (10 cells) and a strong facilitation with a paired-pulse ratio (PPR) of 1.8 ± 0.2 in response to paired-pulses (100 µs, 50 ms interval, *Figure 1B*), typical of CA1 PC synapses (*Lacaille et al., 1987*). Application of baclofen (10 µM) decreased the mean EPSC amplitude by 84% to 12.2 ± 2.2 pA (t$_{(d.f.19)}$=3.96, p=0.001, Holm-Sidak test), but increased the PPR to 2.8 ± 0.5 (t$_{(d.f.25)}$=2.94, p=0.025, Holm-Sidak test, *Figure 1C,D*)

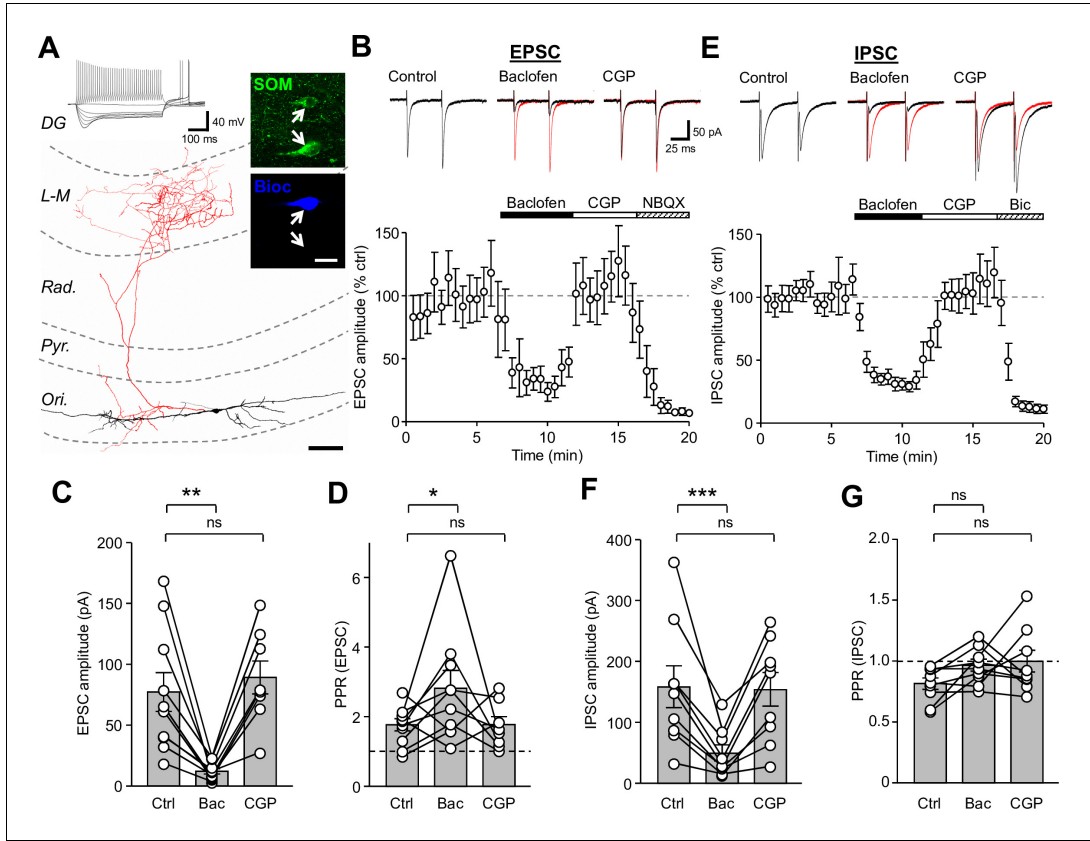

**Figure 1.** Synaptic inputs onto SOM-INs are strongly inhibited by presynaptic GABA$_B$R activation. (**A**) A reconstruction of a SOM-IN showing soma and dendrites (black) and axon (red) with respect to the layers of the CA1 area. Inset (top left), intrinsic physiological response to current injections (500 ms duration; −250 pA to 250 pA). Inset (right), immunoreactivity for SOM (green) in the same biocytin-filled cell (bioc, blue). The arrows indicate SOM immunoreactivity in somata of the biocytin-filled (upper) and non-filled (lower) INs. Scale bar: 20 µm. (**B**, upper) EPSCs elicited in a SOM-IN by *alveus* stimulation in the presence of 10 µM bicuculline, during control (left), baclofen (10 µM, middle, control traces are overlain in red) and CGP-55,845 (CGP, 5 µM, right) bath application. (B, lower) Time-course plot of the mean EPSC amplitude in 10 SOM-INs during control, and sequential bath application of baclofen, CGP and NBQX (10 µM). (**C,D**) Summary bar graphs of EPSC amplitudes and PPRs under control conditions (Ctrl) and during baclofen and subsequent CGP application from 9 SOM-INs. (**E**) Monosynaptic IPSCs evoked by stimulation of *str. oriens* in the presence of NBQX and APV (50 µM, top) and time-course plot of the mean IPSC amplitude from 9 SOM-INs (bottom) during control, and sequential bath application of baclofen, CGP and bicuculline (Bic, 10 µM). (**F,G**) Summary bar graphs of IPSC amplitudes and PPR from 9 SOM-INs. Connected circles correspond to data obtained from a single SOM-IN in the different conditions. Statistics shown: ns – p>0.05, * – p<0.05, ** – p<0.01, *** - p<0.001; all from repeated measures ANOVA with Holm-Sidak post-tests. Abbreviations: Ori - *str. oriens*, Pyr - *str. pyramidale*, Rad - *str. radiatum*, L-M - *str. lacunosum-moleculare*, DG – dentate gyrus.

The online version of this article includes the following figure supplement(s) for figure 1:

**Figure supplement 1.** GABA$_B$R-mediated inhibition of EPSCs in SOM-INs in the mouse hippocampus.

indicating a reduction in the glutamate release probability at presynaptic terminals by GABA$_B$R activation. Subsequent CGP (5 µM) application to the bath in a subset of experiments (8 cells) recovered the EPSCs to baseline levels with a mean amplitude of 89.7 ± 13.7 pA ($t_{(d.f.25)}$=0.68, p=0.502, Holm-Sidak test) and PPR of 1.8 ± 0.2 ($t_{(d.f.19)}$=2.8, p=0.025, Holm-Sidak test, *Figure 1C,D*). The AMPA receptor (AMPAR) antagonist NBQX (10 µM), applied at the end of a subset of these experiments (5 cells) reduced the average EPSC amplitude by 97% to 3.5 ± 1.5 pA ($U_{(d.f. 4)}$=0.0, p=0.0006, Mann-Whitney test), confirming that responses were mediated by glutamatergic synapses.

To explore whether the GABA$_B$R-mediated suppression of feedback CA1 input to SOM-INs was species specific, we performed similar recordings in wild-type mice (5 cells; *Figure 1—figure*

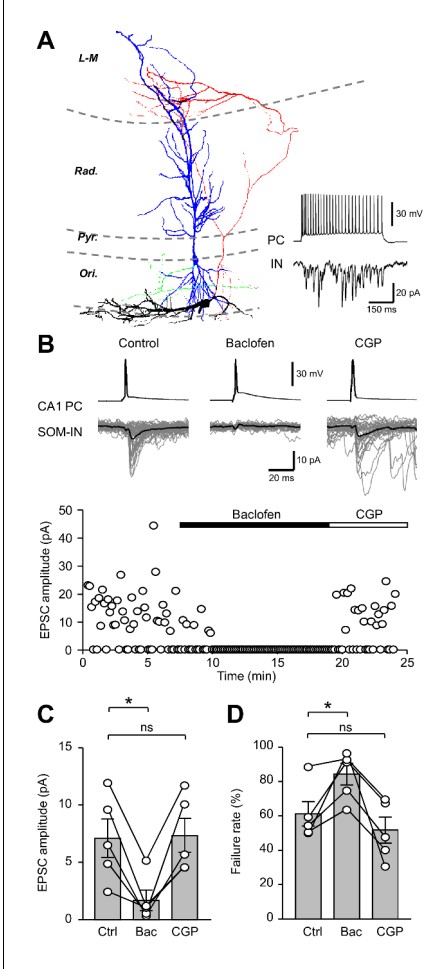

**Figure 2.** Unitary EPSCs from PCs onto SOM-INs are strongly inhibited by presynaptic GABA_BRs. (**A**) Reconstruction of a synaptically coupled CA1 PC (soma/dendrites in blue, axon in green) and SOM-IN pair (soma/dendrites in black, axon in red). Inset (bottom right) shows a train of APs (upper trace) evoked by a depolarizing current applied to the CA1 PC (500 pA, 500 ms) eliciting a shower of EPSCs in the voltage-clamped SOM-IN (lower trace). (**B**) Unitary EPSCs in the SOM-IN (average trace: black line; individual traces: gray lines) produced by single APs evoked by short depolarizing current pulses (1 ms, 1–2 nA) in the CA1 PC. Time course of the unitary EPSC amplitude from the same recording under control conditions and during baclofen (2 µM) and CGP (5 µM) application. (**C**) Summary bar chart of mean EPSC amplitude in 5 CA1 PC/SOM-IN pairs during control conditions (Ctrl), baclofen (Bac) and CGP steady states. (**D**) Summary data for EPSC failure rate recorded under the same conditions. Connected circles correspond to data obtained from a single PC - SOM-IN pair in the different conditions. Statistics shown: ns – p>0.05, * – p<0.05, ** – p<0.01, *** - p<0.001; all from repeated measures ANOVA with Holm-Sidak post-tests.

supplement 1). Under control conditions, stimulation of the *alveus* resulted in an average EPSC amplitude of 62.1 ± 8.5 pA (*Figure 1—figure supplement 1A,B*) which following bath application of a low concentration of baclofen (2 µM) was reduced by 57% to 26.7 ± 4.9 pA ($F_{(d.f.2,4)}$=15.9, p=0.0016, 1-way ANOVA; t = 4.9, p=0.002, Holm-Sidak test; *Figure 1—figure supplement 1B,C*). Subsequent bath application of 5 µM CGP fully reversed the observed inhibition to 101.1% of control (t = 0.01, p=0.99, Holm-Sidak test). These data confirm that strong presynaptic inhibition is present at inputs onto SOM-INs, irrespective of the species tested.

SOM-INs receive a strong GABAergic input from a subset of local CR- and VIP-expressing INs (*Acsády et al., 1996*; *Tyan et al., 2014*). We therefore asked whether GABA_AR-mediated IPSCs in SOM-INs show a similar sensitivity to GABA_BR activation. Paired-pulse extracellular stimulation to the *str. oriens* proximal to these recorded SOM-INs (9 cells) in the presence of AMPAR and NMDAR antagonists (10 µM NBQX and 50 µM APV) gave rise to large monosynaptic IPSCs with an average amplitude of 158.9 ± 34.4 pA and a depressing PPR of 0.82 ± 0.05 (*Figure 1D*). Activation of GABA_BRs by bath application of baclofen (10 µM) reduced the IPSC amplitude by 69% to 49.5 ± 13.2 pA ($t_{(d.f.16)}$=5.07, p=0.0003, Holm-Sidak test), but had minimal and non-significant effect on the PPR (0.97 ± 0.05; $t_{(d.f.16)}$=1.58, p=0.251, Holm-Sidak test; *Figure 1F,G*). CGP (5 µM) application recovered the IPSC amplitude to 97% of control levels (154.5 ± 27.7 pA, $t_{(d.f.16)}$=0.205, p=0.84, Holm-Sidak test; *Figure 1F,G*). Application of bicuculline (10 µM) following CGP recovery inhibited the IPSCs by 94% to 9.9 ± 2.1 pA ($U_{(d.f.9)}$=0, p=0.0006, Mann-Whitney test; *Figure 1D*), confirming their GABA_AR-mediated nature. Thus, our data show that presynaptic GABA_BRs strongly control the strength of both excitatory and inhibitory synaptic inputs onto CA1 SOM-INs.

To examine the strong GABA_BR-mediated inhibition at single excitatory input synapse onto SOM-INs, we next performed paired whole-cell recordings between synaptically coupled CA1 PCs and INs (*Figure 2*). From a total of 42 tested CA1 PC - SOM-IN paired recordings, in 5 pairs (12%) we could observe unitary synaptic coupling in response to a train of presynaptic action potentials (APs) (*Figure 2A*). No IPSCs elicited by SOM-INs were detected in any of the simultaneously recorded PCs.

Single APs produced in the presynaptic CA1 PC resulted in short latency unitary EPSCs with

an average amplitude of 7.1 ± 1.7 pA (range: 2.5–12.0 pA; *Figure 2B,C*) and a failure rate of 61.5 ± 7.2% in SOM-INs (*Figure 2D*). Application of a lower baclofen concentration (2 µM), reduced the mean EPSC amplitude by 74% to 1.6 ± 0.9 pA ($t_{(d.f.\ 8)}$=3.96, p=0.01, Holm-Sidak test; *Figure 2B,C*), consistent with the strong reduction observed with extracellular stimulation ($U_{(d.f.13)}$=24, p=0.945, Mann-Whitney test). Under this condition, the failure rate increased to 84.8 ± 6.3% ($t_{(d.f.8)}$=3.3, p=0.022, Holm-Sidak test; *Figure 2D*), indicating that the release probability was significantly reduced. Subsequent application of 5 µM CGP recovered both the EPSC amplitude ($t_{(d.f.\ 8)}$=0.18, p=0.86, Holm-Sidak test) and the failure rate ($t_{(d.f.\ 8)}$=1.35, p=0.21, Holm-Sidak test, *Figure 2C,D*) to control levels.

These data, thus, confirm that GABA_BRs strongly inhibit excitatory synaptic inputs from local PCs onto SOM-INs, indicating a pivotal role of this receptor in controlling SOM-IN recruitment in this feedback inhibitory microcircuit.

## GABA_BR Subunits are Expressed at Presynaptic Boutons Forming Synapses onto SOM-INs

To investigate the molecular basis of the strong inhibition at excitatory and inhibitory inputs to SOM-INs, we next assessed GABA_BR expression at presynaptic boutons contacting SOM-IN dendrites in *str. oriens/alveus* of CA1. We performed high-resolution quantitative SDS-FRL analysis from perfusion-fixed rat hippocampal sections (*Booker et al., 2017*; *Figure 3—figure supplement 1*). Antibodies used here targeted intracellular epitopes of the proteins of interest - receptors, ion channels, transporters and yellow fluorescent protein (YFP), therefore, they resulted in labeling of the protoplasmic face (P-face), but not the exoplasmic face (E-face), of the replicas (*Figure 3—figure supplement 1A–C*). We first double-labelled replicas for the constitutive GABA_B1 subunit and mGluR1α, which specifically labels CA1 SOM-INs (*Baude et al., 1993*). We consistently observed a high surface expression of GABA_B1 subunits on presynaptic terminals contacting horizontal, sparsely spiny mGluR1α-immunolabelled dendrites in *str. oriens/alveus* (*Figure 3—figure supplement 1A,C*) demonstrating the presence of the receptor at virtually all synapses. To examine the presence of GABA_BRs at glutamatergic and GABAergic presynaptic boutons, we performed double immunolabeling for GABA_B1 with the vesicular glutamate transporter 1 (VGluT1), which selectively labels excitatory terminals in the hippocampus. We consistently observed GABA_B1 labeling at both VGluT1-positive and VGluT1-negative boutons with an apparently higher density of the receptor subunit in VGluT1-negative putative inhibitory terminals (*Figure 3—figure supplement 1D,E*).

To directly assess and quantify the subcellular organization and density of GABA_BRs on dendritic and axonal compartments of SOM-INs, we utilized an approach combining transgenic mouse strategy with the SDS-FRL immunoelectron microscopy. The SOM-Cre mouse line was crossed with the Ai32(RCL-ChR2(H134R)/EYFP) line leading to selective expression of the Channelrhodopsin2(ChR2)-YFP fusion protein in every SOM-IN. As ChR2 renders the fusion protein membrane bound, selective labeling of SOM-IN surface membranes in SDS-FRL replicas can be performed by using an antibody against YFP, thus allowing reliable identification of the INs in SDS-FRL labeling (*Schönherr et al., 2016*; *Trusel et al., 2019*). Indeed, by co-labeling for YFP and mGluR1α, we found that 96% of YFP-positive dendritic shafts in the *str. oriens* (27 out of 28 dendrites) also showed immunoreactivity for this marker glutamate receptor (*Figure 3—figure supplement 2*), confirming that YFP was reliably expressed and detected in SOM-INs in replicas from the transgenic mice. Furthermore, the surface density of immunogold particles for GABA_B1 on YFP-positive dendritic membranes in double-labelled replicas from the transgenic mice was comparable to that previously found in mGluR1α-positive SOM-IN dendrites from rat hippocampal CA1 (54.61 ± 2.52 particles/µm² on YFP-positive mouse dendrites, *Figure 3—figure supplements 3* and 49.1 ± 4.5 particles/µm² in mGluR1α-positive rat dendrites; *Booker et al., 2018*) indicating a convergence between the two rodent species.

In the replicas from transgenic mice, we next performed triple labeling for GABA_B1, YFP and Ca_v2.1 as a marker for presynaptic active zone (*Althof et al., 2015*; *Figure 3*). In good agreement with the findings from rat, we observed an abundant surface expression of the receptor subunit on presynaptic terminals in contact with YFP-positive dendritic shafts of SOM-INs in the *str. oriens* in the mice (*Figure 3A*). Immunoparticles for GABA_B1 were mainly confined to, and distributed non-homogeneously over, the active zones (AZs) of terminals (*Figure 3B,C*), which were recognized by their high density of intramembrane particles on the P-face of the invaginated plasma membrane

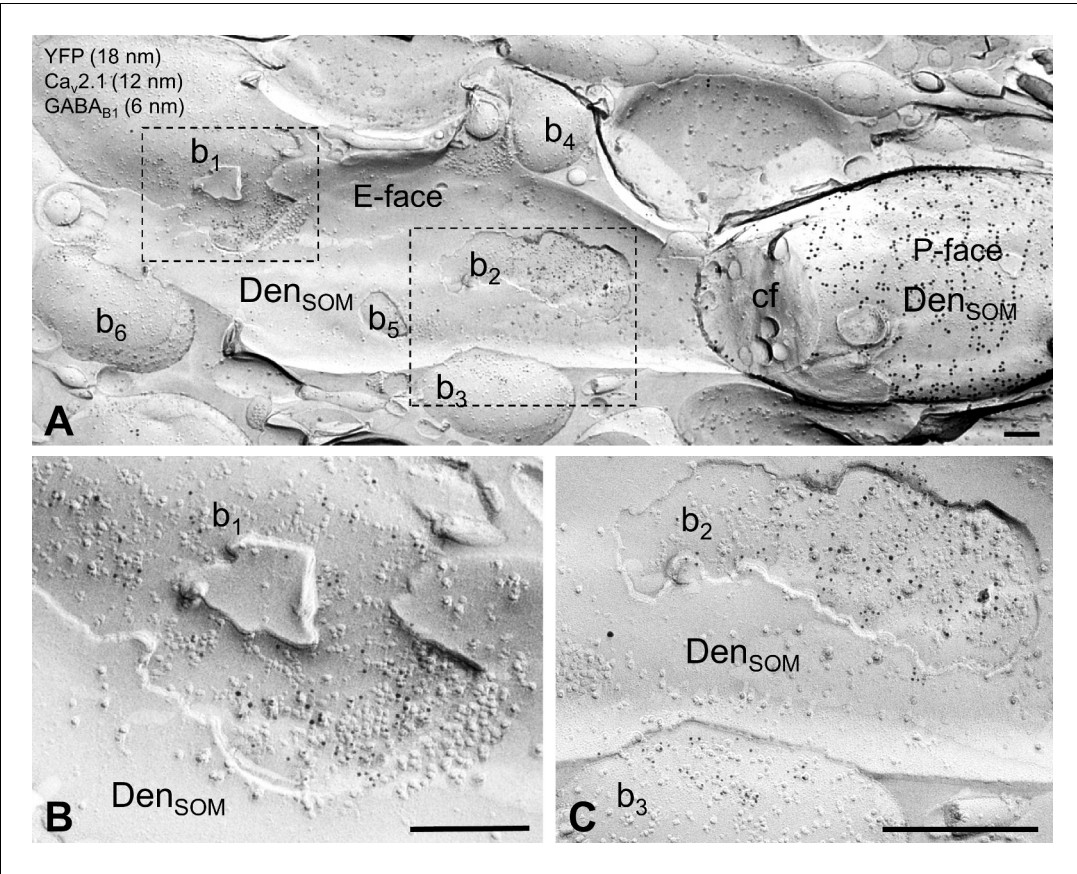

**Figure 3.** Localization of the GABA_B1 subunit to axon terminals contacting SOM-IN dendrites in *str. oriens*. (**A**) Electron micrograph showing the localization of GABA_B1 (6 nm particles) to synaptic membranes, identified by immunoreactivity for Ca_V2.1 (12 nm), and extrasynaptic membrane of axon terminals of putative excitatory (b1) and inhibitory (b2, b3) neurons making synapses onto a dendrite (Den_SOM) of a mouse ChR2-YFP-expressing (18 nm) SOM-IN. (**B,C**) High magnification views of the boxed areas of the SOM-IN dendrite in (**A**). Note that there are three other presumed excitatory boutons (b4-6) contacting dendrite of the IN. Abbreviations: P-face, protoplasmic face; E-face, exoplasmic face; cf, cross-fractured face. Scale bars: 200 nm.

The online version of this article includes the following figure supplement(s) for figure 3:

**Figure supplement 1.** Subcellular distribution of the GABA_B1 subunit in boutons contacting dendritic shafts of *oriens-alveus* INs in the rat hippocampus as revealed by the SDS-FRL method.

**Figure supplement 2.** Colocalization of genetically encoded ChR2-YFP with mGluR1α on putative SOM-IN dendrites.

**Figure supplement 3.** High GABA_B1 subunit surface density on YFP-positive dendritic shafts located in *str. oriens* from SOM-ChR2-YFP mice as revealed by the SDS-FRL method.

and a strong immunolabeling for Ca_V2.1 (*Althof et al., 2015*). Furthermore, receptor subunits were occasionally seen along the extrasynaptic membrane of presynaptic boutons (*Figure 3B,C*).

We next asked to what extent GABA_B1 surface localization is segregated between excitatory and inhibitory presynaptic terminals targeting dendritic shafts of SOM-INs. To address this question, we performed triple-immunolabeling for GABA_B1, YFP and either VGluT1 or vesicular GABA transporter (VGAT), which selectively label boutons of glutamatergic and GABAergic neurons, respectively (*Figure 4*). In these replicas, we consistently observed labeling for GABA_B1 in both populations of terminals, but immunoreactivity for the receptor subunit was significantly higher in VGAT-positive and VGluT1-negative inhibitory terminals ($43.00 \pm 6{,}61$ particles/$\mu m^2$, n = 36) compared to VGAT-negative and VGluT1-positive excitatory presumed PC axon terminals ($17.36 \pm 2.13$ particles/$\mu m^2$, n = 60; $p<0.0001$, Mann-Whitney test; *Figure 4B–D*). These data demonstrate that both GABAergic and glutamatergic terminals forming synapses onto SOM-INs contain large numbers of GABA_BRs,

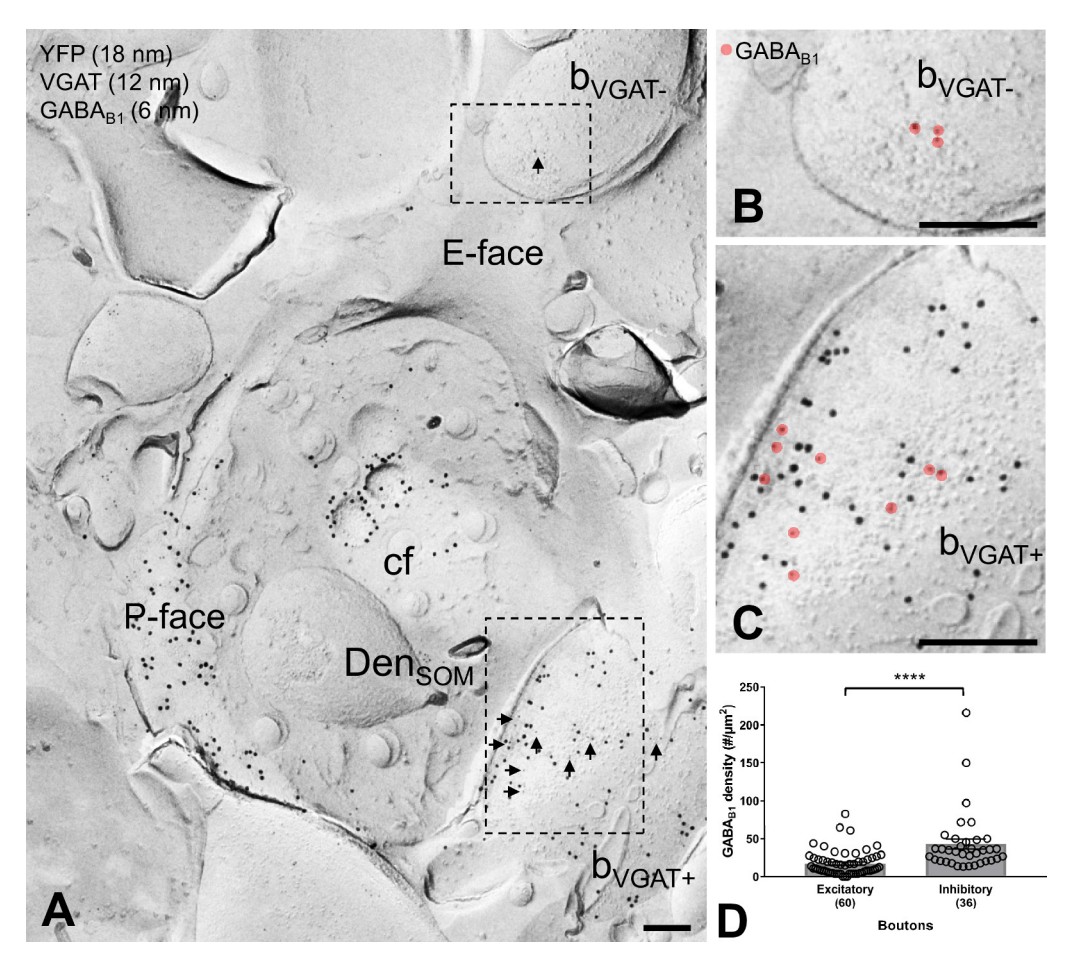

**Figure 4.** GABA$_{B1}$ expression on inhibitory and excitatory axon terminals forming synapses onto SOM-IN dendrites. (**A**) Electron micrographs showing immunogold labeling for GABA$_{B1}$ (6 nm, arrows) on VGAT-immunonegative (b$_{VGAT-}$) and VGAT-immunopositive (b$_{VGAT+}$;12 nm) axon terminals contacting a ChR2-YFP-immunopositive (18 nm) dendritic shaft of a SOM-IN (Den$_{SOM}$). (**B**) Expanded view of the VGAT-immunonegative terminal identified in (**A**), with GABA$_{B1}$ labeling highlighted (red overlay). (**C**) Expanded view of the VGAT-immunopositive terminal identified in (**A**), with GABA$_{B1}$ highlighted. (**D**) Summary bar graph of GABA$_{B1}$ labeling density on excitatory (VGAT- and VGluT1+) and inhibitory (VGAT+ and VGluT1-) axon terminals. Data from individual compartment are shown as open circles with numbers of analyzed terminals in parentheses. Density of GABA$_{B1}$ is significantly higher in inhibitory boutons compared to excitatory terminals (****p<0.0001, Mann-Whitney test). Abbreviations: E-face, exoplasmic face; P-face, protoplasmic face; cf, cross-fractured face. Scale bars: 200 nm.

consistent with electrophysiological findings demonstrating a robust presynaptic inhibition at inhibitory and excitatory inputs, however, with a substantial difference in the surface density between the two bouton populations.

## Activation of GABA$_B$Rs Inhibits GABA Release from SOM-IN Axon Terminals

We next asked whether the output synapses of SOM-INs onto CA1-PCs are inhibited by GABA$_B$ autoreceptors. Given that in acute slice preparations SOM-INs have a very low connection probability to CA1-PCs (*Ali and Thomson, 1998*), we utilized ChR2 activation of SOM-IN axons in SOM-Cre transgenic mice (*Savanthrapadian et al., 2014*; *Yuan et al., 2017*) to examine the synaptic output of SOM-INs (*Figure 5*). For this purpose hippocampal slices were prepared from SOM-Cre transgenic mice injected with rAAVs containing ChR2 and tdTomato between inverted incompatible

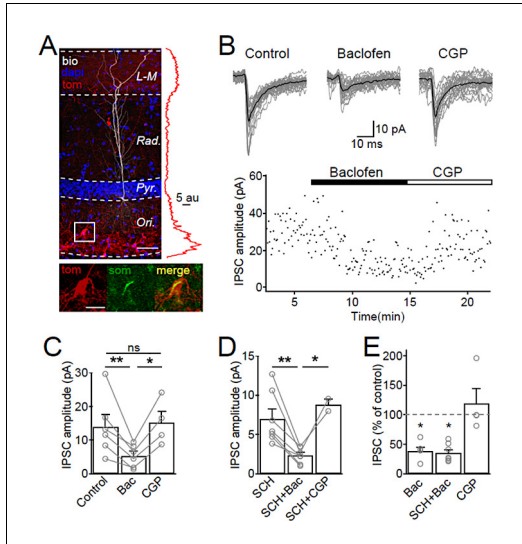

**Figure 5.** SOM-IN output synapses are inhibited by GABA$_B$Rs. (A) Confocal image stack of a recorded and biocytin-filled CA1 PC (white pseudocolor) in a slice from SOM-Cre transgenic mouse transfected with ChR2-tdTomato (tom, red). The slice was counterstained with DAPI (blue) to reveal hippocampal layering, including the cell body layer (Pyr). Inset (right), plot of the fluorescence intensity in the red channel for the ChR2-tdTomato fusion protein across CA1 layers. Inset (bottom), co-localization of tdTomato (tom, red) with SOM immunofluorescence (green). Scale bars: 50 μm (main) and 20 μm (inset). (B, upper) IPSCs recorded in a CA1 PC following photo stimulation of SOM-IN axons in the CA1. Individual sweeps (gray) are overlaid with the averaged traces (in black) for control, baclofen (10 μM) and CGP (5 μM) conditions. (B, lower) Time-course plot of IPSC amplitudes obtained from the CA1 PC during control and sequential bath application of baclofen and CGP. (C) Summary bar chart of mean IPSC amplitudes from 6 CA1 PCs in control, baclofen and CGP conditions. (D) Summary bar chart of IPSC amplitudes from 7 CA1 PCs with the same pharmacological tests, but in the continuous presence of SCH-23,390 (SCH, 10 μM) to block Kir3 channels. (E) Mean IPSC amplitudes normalized to control in the presence of baclofen (6 cells), SCH and baclofen (7 cells), and CGP alone (4 cells). Connected circles correspond to data obtained from single PCs under the different conditions. Statistics shown: ns – p>0.05, *p<0.05, **p<0.01 from Holm-Sidak tests.

The online version of this article includes the following figure supplement(s) for figure 5:

**Figure supplement 1.** Whole-cell baclofen-mediated currents in optogenetic experiments from CA1 PCs of the mouse hippocampus.

tandem loxP sites. This viral strategy allowed spatially restricted expression of ChR2 in the CA1. The expression pattern which was detected by the fluorescence intensity of tdTomato reporter was highest in *str. oriens,* primarily localized to somata and dendrites, and *str. lacunosum-moleculare* showing a more diffuse axonal labeling (*Figure 5A*). At the cellular level this expression was restricted to SOM-INs (*Figure 5A*, inset, bottom).

Optogenetic activation of SOM-IN axons in the *str. lacunosum-moleculare* resulted in temporally-aligned IPSCs in CA1 PCs with an average peak amplitude of 14.0 ± 3.6 pA (6 cells, *Figure 5B,C*). Baclofen application markedly reduced the optically-evoked IPSC amplitudes by 61% to 5.3 ± 1.3 pA ($t_{(d.f.\ 8)}$=4.31, p=0.007, Holm-Sidak test, *Figure 5B,C*) indicating the presence of presynaptic GABA$_B$Rs at SOM-IN output synapses. Subsequent CGP bath-application (4 cells) recovered the IPSC amplitude to 119% of control (15.2 ± 3.4 pA, $t_{(d.f.\ 8)}$=0.12, p=0.90, Holm-Sidak test, *Figure 5B, C*). However, GABA$_B$R activation also produced a strong $I_{WC}$ of 49 ± 15 pA postsynaptically (6 cells, *Figure 5—figure supplement 1A,B*), plausibly due to Kir3 channel opening in CA1 PC dendrites (*Degro et al., 2015*). This postsynaptic conductance may shunt the evoked IPSCs and contribute to their reduced amplitudes during baclofen application. Therefore, we isolated the contribution of presynaptic GABA$_B$Rs in a set of experiments in which we pre-applied the Kir3 channel blocker SCH-23,390 (SCH, 10 μM; *Kuzhikandathil and Oxford, 2002*). Pre-application of SCH reduced the baclofen-induced $I_{WC}$ in CA1 PCs to 19 ± 3 pA (7 cells, $t_{(d.f.\ 11)}$=1, p=0.002, Mann-Whitney test, *Figure 5—figure supplement 1C*). In the presence of SCH, the IPSC elicited by optogenetic activation of

SOM-IN axons had an average amplitude of 7.0 ± 1.3 pA (*Figure 5D*) and tended to be smaller than IPSCs elicited in PCs recorded in the absence of SCH, albeit not significantly so ($U_{(d.f. 11)}$=9, p=0.10, Mann-Whitney test). Baclofen application under this condition inhibited the IPSC by 70% to 2.3 ± 0.4 pA ($t_{(d.f. 7)}$=4.82, p=0.002, Holm-Sidak test, *Figure 5D*). The effect of baclofen was reversed by consecutive CGP application (*Figure 5D*). The baclofen-induced presynaptic inhibition of SOM-INs IPSCs in the presence of SCH was comparable to that of baclofen applied alone ($t_{(d.f. 14)}$=0.2, p=0.84, Holm-Sidak test, *Figure 5E*), indicating that the observed reduction in the IPSC amplitude was not contaminated by a Kir3-mediated postsynaptic shunting effect in the PCs. These data, thus, confirm that GABA$_B$Rs are present in the axon terminals of SOM-INs and strongly inhibit the synaptic output of these INs. Although recent observations indicate that ChR2 expression can interfere with presynaptic GABA$_B$R function (*Liu et al., 2018*), the strong inhibition observed in our study argues against such a scenario under our experimental conditions.

## Presynaptic GABA$_B$Rs are Present at High Density on Axon Terminals of SOM-INs

Our electrophysiological data indicate the presence of functional presynaptic GABA$_B$Rs in axon terminals of SOM-INs. Therefore, we next examined the surface density and subcellular organization of GABA$_{B1}$ in axon terminals of the INs by quantitative SDS-FRL immunoelectron microscopy in replica samples from the *str. lacunosum-moleculare* of CA1 (*Figure 6*) using the same ChR2-YFP-based approach as described above. Triple-immunolabeling of freeze-fracture replicas for YFP, GABA$_{B1}$ and Ca$_V$2.1 demonstrated a high number of GABA$_{B1}$ over the plasma membrane of all analyzed YFP-immunoreactive boutons (*Figure 6A–C*). There was no significant difference in the density of

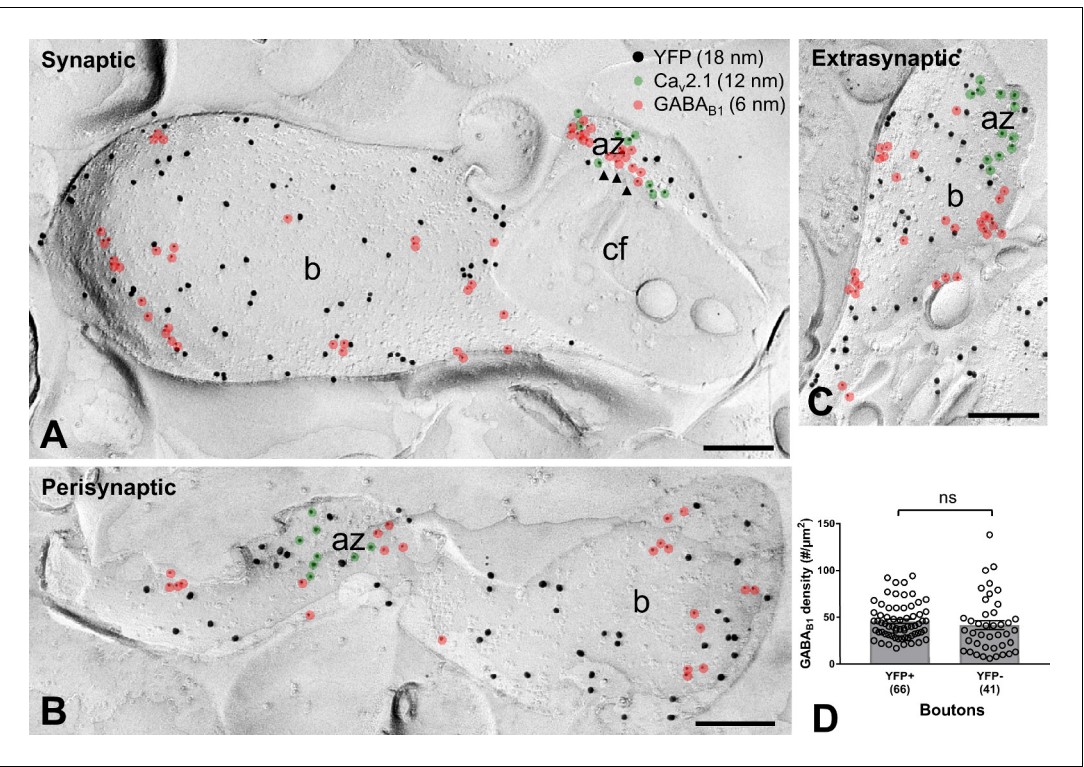

**Figure 6.** GABA$_{B1}$ subunit expression and localization on SOM-IN axon terminals. (**A–C**) Electron micrographs showing GABA$_{B1}$ (6 nm particles, red overlay) localized to the active zone (az) (**A**), identified by immunolabeling for Ca$_V$2.1 (12 nm, green overlay), to the perisynaptic membrane (**B**) and to the extrasynaptic domain (**A–C**) of YFP+ (18 nm) boutons (b) of SOM-INs. Note that extrasynaptic Ca$_V$2.1 channels were not highlighted in (**A**) and (**B**). Arrowheads indicate docked and predocked vesicles on cross-fractured face (cf) of a bouton in (**A**). (**D**) Summary bar chart of the average GABA$_{B1}$ density on YFP+ boutons, compared to nearby YFP- boutons. Data from individual compartment are shown (open circles) with number of analyzed terminals in parentheses. ns – p>0.05. Mann-Whitney test. Scale bars: 200 nm.

immunoparticles for the receptor subunit on YFP-positive terminals (45.57 ± 2.31 particles/μm$^2$, n = 66) of SOM-INs and YFP-negative putative PC boutons (41.56 ± 4.71/μm$^2$, n = 41; p=0.099, Mann-Whitney test) (*Figure 6D*). Furthermore, detailed analysis revealed that 20% and 37% of the terminals possess high number of synaptic (*Figure 6A*) and perisynaptic (*Figure 6B*) receptors, respectively, whereas 43% of them contain exclusively extrasynaptic GABA$_{B1}$ (*Figure 6C*). These data suggest a substantial heterogeneity among SOM-IN axon terminals in terms of ultrastructural localization of presynaptic GABA$_B$Rs.

## Presynaptic GABA$_B$Rs Functionally Uncouple SOM-INs from the Local Microcircuit

SOM-INs participate in synchronized hippocampal network oscillations - in particular theta (4–12 Hz), but also gamma frequency (30–100 Hz) activity (*Gloveli et al., 2005b*; *Hájos et al., 2004*; *Huh et al., 2016*; *Maccaferri and McBain, 1996*). Given the near complete inhibition of excitatory, as well as inhibitory, synaptic inputs onto SOM-INs by presynaptic GABA$_B$Rs, we asked whether this inhibition was able to functionally uncouple SOM-INs from network oscillations (*Figure 7*).

First, we induced slow, theta frequency oscillations in hippocampal slices by bath applying carbachol (50 μM; *Figure 7A*). We recorded extracellular theta activity in 8 slices and in 5 of these slices SOM-INs were recorded simultaneously to the local field potential. The peak power of the oscillatory activity in the field was 27.5 ± 7.0 μV$^2$ at a frequency of 9.8 ± 1.5 Hz (*Figure 7B*). During the oscillations, SOM-INs had an average discharge frequency of 13.0 ± 3.2 Hz with the APs occurring at the rising phase of the theta cycle (*Figure 7C*), consistent with the discharge pattern observed in vivo (*Forro et al., 2015*). Application of 2 μM baclofen resulted in a strong, 78% reduction in the theta peak power to 8.8 ± 3.3 μV$^2$ (t$_{(d.f.8)}$=3.8, p=0.015, Holm-Sidak test; *Figure 7A,B*). The reduction in theta power was complemented by a near-complete loss of SOM-IN discharge (mean frequency of 0.4 ± 0.2 Hz; t$_{(d.f. 8)}$=3.3, p=0.03, Holm-Sidak test, *Figure 7A,D*). Both the theta peak power (21.5 ± 5.0 μV$^2$; t$_{(d.f. 8)}$=1.22, p=0.26, Holm-Sidak test; *Figure 7B*) and the discharge frequency (14.1 ± 4.0 Hz; t$_{(d.f. 8)}$=0.12, p=0.91, Holm-Sidak test; *Figure 7D*) were fully reversed by subsequent bath application of 5 μM CGP, confirming that the silencing of SOM-INs was mediated by GABA$_B$Rs. Recordings from CA1 PCs (4 cells) showed that their peak discharge frequency (control: 3.9 ± 1.5 Hz) was markedly reduced, but not abolished during baclofen application (1.4 ± 0.9 Hz; t$_{(d.f.3)}$=3.7, p=0.025, Holm-Sidak test), reflecting that the effect of GABA$_B$R activation had a network wide impact. The discharge of PCs, similar to that of the INs, was fully recovered by CGP application (4.1 ± 1.4 Hz; t$_{(d.f.3)}$=1.21, p=0.258, Holm-Sidak test).

SOM-INs are also recruited to gamma oscillations (*Tort et al., 2007*), therefore, we next tested the effects of presynaptic GABA$_B$R activation on SOM-IN firing during this faster activity pattern. Puff application of kainate (2 mM, 100 ms) to *str. radiatum* at a distance of ~100 μm from the recorded SOM-IN, resulted in large amplitude fast gamma oscillations in the local field potential recorded from the *str. radiatum*, which lasted 5–10 s (*Figure 7E*). These oscillations had a peak power of 52.6 ± 14.3 μV$^2$ at 61.1 ± 4.4 Hz (*Figure 7F*). During the kainate-induced gamma oscillations SOM-INs fired at 7.7 ± 2.0 Hz (6 cells, *Figure 7H*), comparable to that observed during theta oscillations (U$_{(d.f. 9)}$=7, p=0.16, Mann-Whitney test). Individual APs in SOM-INs preferentially occurred at the ascending phase of gamma oscillation, on average −15 ± 10° before the peak of the oscillatory cycle (*Figure 7G,I*) indicating a high correlation and tight phase relationship to local network activity, despite their low discharge frequency. Bath application of 2 μM baclofen reduced the peak power of gamma oscillations by ~45% to 28.7 ± 12.0 μV$^2$ (F$_{(d.f. 2,10)}$=5.13, p=0.029, 1-way ANOVA; *Figure 7F*) with minimal effect on the frequency (65.5 ± 3.8 Hz, t$_{(d.f.10)}$=1.1, p=0.51, Holm-Sidak test). Baclofen application also produced a near-complete loss of APs in SOM-INs (6 cells) reducing the discharge frequency to 0.7 ± 0.2 Hz (t$_{(d.f.5)}$=3.6, p=0.04, Holm-Sidak test, *Figure 7H*). In contrast, baclofen application only marginally affected the discharge frequency of CA1 PCs under the same conditions (3 cells) reducing it from 9.6 ± 2.2 Hz to 7.9 ± 3.9 Hz (t$_{(d.f.2)}$=0.7, p=0.80, Holm-Sidak test).

Given the near-complete loss of SOM-IN discharge, it was not possible to assess whether presynaptic GABA$_B$R activation led to a breakdown of SOM-IN phase coupling to the oscillations. Therefore, we applied a depolarizing current to SOM-INs to promote their firing in the presence of baclofen. When SOM-INs were held near their rheobase current (106 ± 31 pA; *Figure 7I*, middle) in the presence of 2 μM baclofen, they fired at low frequency (2.9 ± 0.6 Hz). Under these conditions,

the APs showed no preference to the phase of gamma activity ($F_{(d.f.\ 18,57)}$=1.8, p=0.048, 2-way repeated-measures ANOVA). The discharge frequency (11.6 ± 3.2 Hz, p=0.22, Holm-Sidak test, compared to control conditions; *Figure 7H*) as well as the phase preference (25 ± 22°, *Figure 7I*, right) recovered during subsequent application of CGP (5 µM, without bias current).

To confirm that the strong inhibition of SOM-IN discharge during gamma oscillations was not species specific, we performed the same experiments in acute hippocampal slices from mice (*Figure 7—figure supplement 1*). Puff application of kainate to *str. radiatum* produced reliable gamma oscillations with a peak power of 52.8 ± 19.2 µV$^2$ at a frequency of 44.9 ± 9.4 Hz (4 slices, *Figure 7—figure supplement 1A,B*). Bath application of 2 µM baclofen reduced the peak power to 36.6 ± 13.2 µV$^2$ ($F_{(d.f.\ 2,7)}$=38.4, p=0.0002 1-way ANOVA; p=0.001, Holm-Sidak test; *Figure 7—figure supplement 1B*). Simultaneous intracellular recording from SOM-INs during the kainate-puff-induced gamma oscillation revealed a discharge in the theta range with a mean frequency of 7.9 ± 0.9 Hz under control conditions. Similar to the observations in rat slices, application of baclofen also strongly attenuated discharge to 1.6 ± 0.7 Hz in mouse SOM-INs ($t_{(d.f.\ 7)}$=6.0, p=0.0.001, Holm-Sidak test; *Figure 7—figure supplement 1C*). The oscillatory power, as well as the spike rate recovered with subsequent 5 µM CGP bath-application in 2 SOM-INs tested (*Figure 7—figure supplement 1B,C*). These results are consistent with an uncoupling of SOM-INs from the network by presynaptic GABA$_B$R activation in both rats and mice.

To demonstrate that the reduction in spiking was due to a local, not global reduction in synaptic transmission and excitability (as caused by bath application of baclofen), we performed experiments in which we focally applied baclofen (2 mM) via a second puff electrode in close proximity to the SOM-IN somata (*Figure 7—figure supplement 2*). Under control conditions, the peak power of gamma oscillations induced by the kainate puff to the *str. radiatum* was 48.8 ± 11.2 µV$^2$ at 37.0 ± 2.4 Hz in these experiments (*Figure 7—figure supplement 2A,B*). Focal puff application of baclofen to *str. oriens* had no significant effect on the peak power of gamma activity in 8 slices tested (37.6 ± 9.3 µV$^2$; $F_{(d.f.\ 2,16)}$=0.33, p=0.72; 1-way ANOVA; p=0.82, Holm-Sidak test; *Figure 7—figure supplement 2B*). Despite no change in peak gamma power, the focal baclofen application produced a near complete abolition of SOM-IN discharge from 10.2 ± 3.7 Hz under control conditions to 0.1 ± 0.1 Hz during the baclofen puff in 6 SOM-INs ($F_{(d.f.\ 2,13)}$=4.7, p=0.028 1-way ANOVA; p=0.032, Holm-Sidak test; *Figure 7—figure supplement 2C*). The reduction in SOM-IN firing was reversed to 7.8 ± 1.7 Hz by bath application of 5 µM CGP during the baclofen puff in 4 cells (Compared to control: p=0.54, Holm-Sidak test) confirming the GABA$_B$R dependence of this effect.

In summary, GABA$_B$R-mediated inhibition is very strong at synaptic inputs to SOM-INs and can functionally uncouple them from the local network, silencing these INs during oscillatory activity.

## Discussion

In the current study we show that excitatory and inhibitory axon terminals converging onto SOM-INs express high levels of GABA$_B$Rs and activation of the receptors leads to strong inhibition of transmission at these synapses. The synaptic output of SOM-INs onto CA1 PC dendrites is similarly inhibited by GABA$_B$R activation. We find that the strong presynaptic GABA$_B$R inhibition observed is sufficient to silence SOM-INs during in vitro theta and gamma oscillations. Combined, these data provide strong evidence that SOM-INs can be silenced and uncoupled from hippocampal microcircuits by GABA$_B$R-mediated presynaptic inhibition during co-ordinated network activity when extracellular GABA levels surge (*Scanziani, 2000*).

### Presynaptic GABA$_B$R Activation Suppresses Synaptic Inputs onto SOM-IN

The primary excitatory synaptic input to SOM-INs is from CA1 PCs, making these INs a major feedback inhibitory element in cortical circuits (*Ali and Thomson, 1998*; *Lacaille et al., 1987*; *Shigemoto et al., 1996*). Previous studies have demonstrated that PC synapses onto SOM-INs produce small depolarization, with high failure rates, and strong frequency dependent facilitation, indicative of a low initial release probability synapse (*Pala and Petersen, 2015*; *Silberberg and Markram, 2007*; *Urban-Ciecko et al., 2018*). This low basal synaptic transmission onto SOM-INs has been suggested to arise from multiple presynaptic modulatory pathways, including group II and III mGluRs (*Shigemoto et al., 1996*; *Losonczy et al., 2003*) and 5-HT$_{1A}$ receptors in the hippocampus

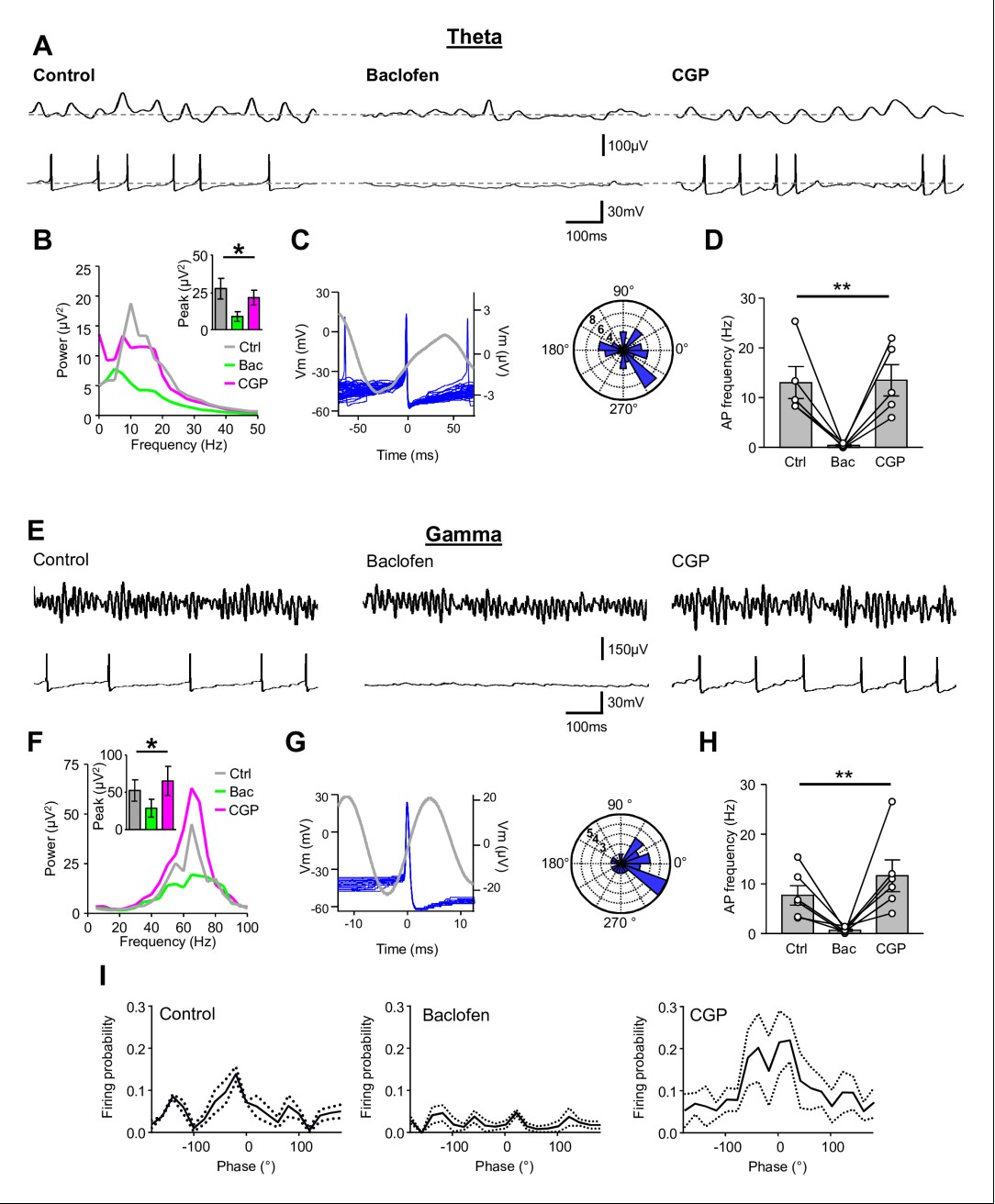

**Figure 7.** Presynaptic GABA$_B$R-mediated inhibition functionally uncouples SOM-INs from the local network. (A) Dual extracellular field (upper) and whole-cell (lower) recordings from a CA1 SOM-IN in longitudinal acute hippocampal slice during theta frequency oscillations induced by bath application of 50 µM carbachol. Representative traces are shown in control conditions (left), and during application of baclofen (2 µM, middle), or subsequent application of CGP (5 µM, right). (B) Spectral analysis of the field oscillations under control conditions (grayscale), during baclofen (Bac, green) and subsequent CGP (magenta) application (7 slices). Inset, bar chart of the average peak power for the three pharmacological conditions. (C) Spike triggered average of the field potential (gray trace)± 75 ms from the AP in a SOM-IN (blue, superimposed; left) illustrating the firing preference at the rising phase of the oscillations. Inset (right), polar plot of the phase preference of SOM-IN discharge with respect to the oscillatory cycle. (D) Summary bar chart of the average discharge frequency of SOM-INs (7 cells) during theta frequency activity in control conditions and in the presence of baclofen or CGP. (E–H) Corresponding data for gamma activity induced by 2 mM kainate puff application to *str. radiatum* of CA1 in horizontal hippocampal slices. (I) Mean discharge probability of SOM-INs plotted as a function of the phase of gamma under control conditions (left), in the presence of baclofen (middle, with a bias current applied to hold the membrane

*Figure 7 continued on next page*

*Figure 7 continued*
potential of the SOM-IN close to firing threshold), and in the presence of CGP (right, without bias current). Statistics shown: ns – p>0.05, ** - p<0.01, derived from repeated-measures ANOVA.
The online version of this article includes the following figure supplement(s) for figure 7:

**Figure supplement 1.** SOM-IN spiking is abolished by GABA$_B$R activation in the mouse hippocampus.
**Figure supplement 2.** Abolished SOM-IN spiking is due to GABA$_B$Rs activation in CA1 *str. oriens* of the mouse hippocampus.

(*Böhm et al., 2015*). In contrast, nicotinic acetylcholine receptors have been recently shown to boost neurotransmission at the input synapses of SOM-INs in the neocortex (*Urban-Ciecko et al., 2018*). Our results reveal that GABA$_B$Rs are also abundant at presynaptic terminals onto SOM-INs, at both synaptic and extrasynaptic membranes. While we did not observe evidence for tonic presynaptic GABA$_B$R activation under our experimental conditions, agonist activation consistently produced inhibition of the glutamatergic input onto SOM-INs, sufficient to strongly inhibit monosynaptic and unitary EPSCs with concomitant changes in failure rate and PPR, indicative of a presynaptic locus of action. Plausibly, during periods of high network activity, as observed in vivo, GABA spill-over from local inhibitory synapses may readily activate GABA$_B$Rs (*Scanziani, 2000*; *Oláh et al., 2009*) in a dynamic, state-dependent manner. However, higher ambient levels of GABA could also produce a tonic level of GABA$_B$R activation, contributing to the weak glutamatergic transmission from PCs to SOM-IN, through heterosynaptic receptor activation (*Urban-Ciecko et al., 2015*).

Similar to glutamatergic synapses, we found a consistent, high level of surface expression of GABA$_B$Rs and a strong presynaptic inhibition at GABAergic synapses onto SOM-INs. Previous studies indicate that VIP/CR-containing INs account for 70% of all inhibitory synapses onto SOM-INs (*Acsády et al., 1996*; *Tyan et al., 2014*). As such, we infer that the output of VIP/CR INs is strongly controlled by GABA$_B$ autoreceptors. While the precise temporal pattern of this inhibitory synaptic input to SOM-INs is not known, the strong suppression of transmission during GABA$_B$R activation is likely to contribute to the reduced phase relationship and uncoupling of these INs from the local network.

## GABA$_B$R Expression and Function at the Inhibitory Synaptic Output of SOM-IN

The major SOM-IN output is onto the distal apical dendrites of PCs, and also other INs in hippocampal CA areas (*Katona et al., 1999*). Their postsynaptic effects are primarily mediated by GABA$_A$R (*Huh et al., 2016*; *Maccaferri et al., 2000*), but slow GABA$_B$R effects have also been observed in hippocampal and neocortical PCs (*Huh et al., 2016*; *Maccaferri et al., 2000*; *Urban-Ciecko et al., 2015*; *Nichol et al., 2018*), indicating that in addition to neurogliaform cells, SOM-INs may contribute to GABA volume transmission (*Tamás et al., 2003*; *Price et al., 2008*; *Oláh et al., 2009*). Indeed, SOM-INs can produce strong GABA$_B$R-mediated heterosynaptic inhibition at excitatory synapses onto cortical PCs in line with a role in volume transmission (*Urban-Ciecko et al., 2015*). The observed high density of GABA$_B$Rs in SOM-IN axon terminals and the strong depression of their inhibitory output during receptor activation in our optogenetic experiments demonstrate that GABA$_B$ autoreceptors dynamically regulate GABA release from these synapses. However, about half (57%) of SOM-IN boutons possessed GABA$_B$Rs localized within or in direct apposition to the presynaptic active zone, whereas essentially all terminals contained extrasynaptic GABA$_B$R - a subcellular distribution pattern observed at other cortical synapses (*Kulik et al., 2003*). This suggests that a functional dichotomy may exist where most SOM-IN synapses are regulated by ambient GABA levels, with approximately half of synapses also inhibited by GABA$_B$ autoreceptors. The cellular sources of GABA involved in heterosynaptic inhibition at the output of SOM-INs is likely to be distinct from those at the input synapses, however their effects could converge to uncouple the INs from the network under specific conditions. Alternatively, this distribution pattern may demonstrate a high degree of surface dynamics of presynaptic GABA$_B$Rs reflecting the activity level of the INs. This observed heterogeneity of GABA$_B$R compartmentalization and its functional impact in SOM-IN outputs should be the subject of further studies.

## Role of SOM-INs and GABA$_B$Rs in Network Activity

SOM-INs are well known to participate in oscillatory activity intrinsic to cortical circuits (*Gloveli et al., 2005a*; *Hájos et al., 2004*; *Klausberger et al., 2003*; *Maccaferri and McBain, 1996*; *Müller and Remy, 2014*; *Pangalos et al., 2013*; *Tort et al., 2007*). In good agreement with previous findings, both in vitro and in vivo (*Gloveli et al., 2005b*; *Huh et al., 2016*; *Klausberger et al., 2003*; *Varga et al., 2012*), we observed that hippocampal SOM-INs discharge phase-locked to the field oscillation during both theta and gamma rhythms. These two oscillatory patterns in vitro were sensitive to low, presynaptic-selective concentrations of baclofen (*Dugladze et al., 2013*; *Vigot et al., 2006*). This result is complementary to previous findings that GABA$_B$R antagonists increase the oscillatory power in these two frequency bands (*Johnson et al., 2017*; *Leung and Shen, 2007*) and plausibly reflects circuit-wide disinhibition of neuronal excitability and transmission, enhancing phasic modulation of the network by fast inhibition. However, the sensitivity of the two oscillatory patterns to GABA$_B$R activation was differential in our experiments: while theta rhythm was almost completely abolished, gamma activity persisted albeit with a reduced power. The high sensitivity of theta activity to baclofen may reflect an inherent lability of in vitro theta oscillations. Indeed, only a few publications have reported reliable theta oscillations in ex vivo preparations (*Fellous and Sejnowski, 2000*; *Gloveli et al., 2005a*; *Goutagny et al., 2009*) which might be due to the strong reliance of an intact connection from the entorhinal cortex or the septum (*Buzsáki, 2002*). However, given the proposed central role of SOM-INs in the generation of theta activity (*Forro et al., 2015*; *Gloveli et al., 2005a*; *Hájos et al., 2004*; *Klausberger et al., 2003*; *Sekulić and Skinner, 2017*), synaptic uncoupling of the INs by presynaptic GABA$_B$Rs may also be causally related to the abolished theta activity. In this context the persistence of gamma oscillations could be explained by their microcircuit mechanisms involving other IN types, in particular fast-spiking basket cells (*Bartos et al., 2002*; *Bartos et al., 2007*; *Gulyás et al., 2010*).

Pharmacological models of gamma do not fully reflect the nuances of in vivo activity, and as such rarely does gamma exist as a prolonged 5–10 s barrage, but rather occurs co-generated with theta as nested short bursts both in the hippocampus and in the neocortex (*Bragin et al., 1995*; *Johnson et al., 2017*; *Strüber et al., 2017*). As SOM-INs are recruited to these oscillations they rhythmically discharge and provide inhibition to the apical dendrites of PCs. In the CA1, this inhibition gates the entorhinal inputs to PCs (*Leão et al., 2012*) and has an essential role in learning and memory processes (*Lovett-Barron et al., 2014*). Our results show that presynaptic GABA$_B$Rs are strategically positioned at both input and output synapses of SOM-INs in this generic cortical microcircuit motif, and can exert strong suppression on transmission. As SOM-IN recruitment and their precisely-timed discharge are primarily defined by their excitatory input from PCs (*Huh et al., 2016*; *Urban-Ciecko et al., 2018*) the strong suppression by GABA$_B$Rs alone at this synapse, may be sufficient to uncouple and silence SOM-INs. However, the parallel suppression of GABAergic input synapses will leave the SOM-INs with minimal phasic synaptic control from the active network. Silencing of SOM-IN firing allows for a breakthrough of multi-sensory inputs from the temporoammonic pathway onto hippocampal PCs, leading to strengthened inhibition and suppressed plasticity at intrahippocampal connections (*Leão et al., 2012*). The differential sensitivity of gamma and theta oscillations further suggests that the uncoupling of SOM-INs from the network by GABA$_B$R activation may also modify the balance of these activity patterns. GABA$_B$R-mediated suppression of SOM-IN output provides a convergent mechanism for the modulation of this feedback microcircuit. However, in view of the wide spatial separation of input and output synapses in *str. oriens* and str. *lacunosum-moleculare*, ambient GABA levels, and thus GABA$_B$R activation, likely displays both temporal and pathway specific separation.

Taken together, the data we present suggest that the activity of SOM-INs, a major feedback element in cortical circuits, is strongly regulated by presynaptic GABA$_B$Rs at their input and output synapses. The suppression of synaptic transmission serves to functionally silence and uncouple SOM-INs from ongoing coordinated network activity. These actions of GABA$_B$Rs can fine tune inhibition/excitation balance in a compartment specific-manner and thereby tightly control the routing of information between intra- and extrahippocampal pathways.

# Materials and methods

**Key resources table**

| Reagent type (species) or resource | Designation | Source or reference | Identifiers | Additional information |
|---|---|---|---|---|
| Genetic reagent (R. norvegicus) | Wild-type rat | Charles River | Crl:WI | Rats used for immunoelectron microscopy |
| Genetic reagent (R. norvegicus) | vGAT-YFP rat | PMID: 17517679 | | Rats used for acute slice experiments |
| Genetic reagent (M. musculus) | Wild-type mouse | Charles River | C57/Bl6J CRL | Mice used for acute slice experiments |
| Genetic reagent (M. musculus) | Som-Cre mouse | Jackson Laboratory; 013044 | Ssttm2.1(cre)Zjh/J | |
| Genetic reagent (M. musculus) | Ai32 mouse | Jackson Laboratory; 024109 | RCL-ChR2 (H134R)/EYFP | |
| Antibody | Anti-GFP | Aves labs, USA | Aves labs: GFP-1010 | chicken polyclonal (1:25000) |
| Antibody | Anti- Cav2.1 | Frontier Institute, JAPAN | Frontier Institute: VDCCa1A-GP-Af810; RRID:AB_2571851 | guinea pig polyclonal (1:50) |
| Antibody | Anti- VGAT | Frontier Institute, JAPAN | Frontier Institute. VGAT-GP-Af1000; RRID:AB_2571624 | guinea pig polyclonal (1:44) |
| Antibody | Anti-VGLUT1 | Frontier Institute, JAPAN | Frontier Institute: VGluT1-Go-Af310; RRID:AB_2571617 | goat polyclonal (1:400) |
| Antibody | Anti-mGluR1$\alpha$ | Frontier Institute, JAPAN | Frontier Institute: mGluR1a-GP-Af660; RRID:AB_2571801 | guinea pig polyclonal (1:500) |
| Antibody | Anti-Somatostatin | Peninsula Laboratories, USA | Peninsula Laboratories: T-4103.0050 | rabbit polyclonal (1:2000) |
| Antibody | Anti-GABA$_{B1}$ (B17) | PMID: 11849296 | | Rabbit polyclonal; against aa 901–960; (1:50) |
| Antibody | Anti-GABA$_{B1}$ (A25) | PMID: 16427742 | | Rabbit polyclonal; against aa 857–960; (1:200) |
| Antibody | 6 nm anti-Rabbit IgG (secondary) | Jackson ImmunoResearch Europe, UK | Jackson ImmunoResearch E:711-195-152 | (1:30) |
| Antibody | 12 nm anti-Guinea pig (secondary) | Jackson ImmunoResearch Europe, UK | Jackson ImmunoResearch E:706-205-148 | (1:30) |
| Antibody | 12 nm anti-Goat (secondary) | Jackson ImmunoResearch Europe, UK | Jackson ImmunoResearch E:705-205-147 | (1:30) |
| Antibody | 18 nm anti-Chicken (secondary) | Jackson ImmunoResearch Europe, UK | Jackson ImmunoResearch E:703-215-155 | (1:30) |
| Antibody | 3 nm anti-Guinea pig (secondary) | Nanopartz, USA | Nanopartz:CA11-3-FCDAGG-DIH-50–1 | (1:800) |

*Continued on next page*

*Continued*

| Reagent type (species) or resource | Designation | Source or reference | Identifiers | Additional information |
|---|---|---|---|---|
| Antibody | Alexa Fluor 405,488,546 Anti-Rabbit (secondaries) | Invitrogen, UK | | (1:500) |
| Other | Alexa Fluor 647 Streptavidin | Invitrogen, UK | Invitrogen:S21374 | (1:500) |
| Chemical compound, drug | NBQX | Abcam Biochemicals, UK | ab120046 | |
| Chemical compound, drug | DL-AP5 | Abcam Biochemicals, UK | ab120271 | |
| Chemical compound, drug | Bicuculline | Abcam Biochemicals, UK | ab120108 | |
| Chemical compound, drug | Gabazine | Abcam Biochemicals, UK | ab120042 | |
| Chemical compound, drug | R-Baclofen | Abcam Biochemicals, UK | ab120325 | |
| Chemical compound, drug | CGP-55845 | Abcam Biochemicals, UK | ab120337 | |
| Chemical compound, drug | SCH23,390 | Tocris, UK | 0925/10 | |
| Chemical compound, drug | Kainate | Tocris, UK | 0222/1 | |
| Chemical compound, drug | Carbcahol | Abcam Biochemicals, UK | ab141354 | |
| Software, algorithm | GraphPad Prism | GraphPad Software, USA | | |
| Software, algorithm | WinWCP | University of Strathclyde, UK | | http://spider.science.strath.ac.uk/sipbs/software_ses.htm |
| Software, algorithm | pClamp | Axon Instruments, USA | | |
| Software, algorithm | MATLAB | Mathworks, USA | | |
| Software, algorithm | StimFit | PMID: 24600389 | | |
| Software, algorithm | Fiji | | | http://fiji.sc |

## Animals

Electrophysiological experiments were performed in acute slices prepared from 17 to 26 day-old wild-type and transgenic Wistar rats expressing Venus/yellow fluorescence protein (YFP) under the vesicular GABA transporter (VGAT) promoter (*Uematsu et al., 2008*) or 25–30 day old wild-type mice (C57/Bl6J$_{CRL}$). For optogenetic experiments, 10–12 week-old SOM-Cre mice (Jackson Laboratories; Ssttm2.1(cre)Zjh/J; *Taniguchi et al., 2011*) were bilaterally injected with rAAVs containing Channelrhodopsin2 (ChR2) and tdTomato (tdTom) coding regions between inverted incompatible tandem loxP sites into the hippocampal CA1 area (coordinates from Bregma: 2 mm, 2 mm; 1.4 mm;

3 µl volume, 3 mins). Optogenetic experiments were performed 2–4 weeks following viral injection. Electron microscopy was performed on either 8-week-old male Wistar rats or 6-week-old male SOM-cre mice crossed with Ai32 (RCL-ChR2(H134R)/EYFP) transgenic mice (The Jackson Laboratory, stock number: 024109, Maine). Care and handling of the animals prior to and during the experimental procedures followed European Union and national regulations (German Animal Welfare Act; ASPA, United Kingdom Home Office) and all experiments were performed in accordance with institutional guidelines (Charité - Universitätmedizin Berlin; University of Freiburg, Freiburg, Germany), with permissions from local authorities (LaGeSo, Berlin, T-0215/11 LaGeSo; Freiburg, X14/11H and 35–9185.81/G-19/59).

## Acute slice preparation

Acute hippocampal slices were prepared and recordings performed as previously described *Booker et al. (2014)*. Briefly, rodents were decapitated (either directly or following cervical dislocation) and their brain rapidly dissected and chilled in semi-frozen carbogenated (95% $O_2$/5% $CO_2$) sucrose-substituted artificial cerebrospinal fluid (sucrose-ACSF, in mM: 87 NaCl, 2.5 KCl, 25 $NaHCO_3$, 1.25 $NaH_2PO_4$, 25 glucose, 75 sucrose, 7 $MgCl_2$, 0.5 $CaCl_2$, 1 Na-Pyruvate, 1 Na-Ascorbate). Transverse hippocampal slices (300 or 400 µm thick) were cut on a vibratome (VT1200s, Leica, Germany) and stored submerged in sucrose-ACSF warmed to 35°C for at least 30 min and subsequently at RT. For recording of network oscillations, slices were stored in a liquid/gas interface chamber, which was perfused with normal ACSF (in mM: 125 NaCl, 2.5 KCl, 25 $NaHCO_3$, 1.25 $NaH_2PO_4$, 25 glucose, 1 $MgCl_2$, 2 $CaCl_2$, 1 Na-Pyruvate, 1 Na-Ascorbate, pH 7.4) at 30–32°C from slicing until recording in order to maintain active oscillatory activity (*Hájos et al., 2009*).

## Whole-cell patch-clamp recordings

For electrophysiological recordings, slices (300 µm thick) were placed in a submerged recording chamber, perfused with carbogenated ACSF at 10–12 ml/min and maintained at near physiological temperatures (32 ± 1°C) using an inline heater (Supertech Instruments, Pécs, Hungary). Slices were viewed under infrared Köhler illumination by means of an upright microscope (BX-50 or BX-51, Olympus, Hamburg, Germany or SliceScope, Scientifica, UK) with a 40x water-immersion objective lens (N.A. 0.8). Whole-cell patch-clamp recordings were accomplished using either an AxoPatch 200B or Multiclamp 700B amplifier (Molecular Devices, USA) and recording pipettes pulled from borosilicate glass capillaries (2 mm outer/1 mm inner diameter, Hilgenberg, Germany) on a horizontal electrode puller (P-97, Sutter Instruments, CA, USA). Pipettes were filled with intracellular solution (in mM: 120 K-Gluc, 20 KCl, 2 $MgCl_2$, 10 EGTA, 10 HEPES, 2 $Na_2$-ATP, 0.3 $Na_2$-GTP, 1 $Na_2$-Creatinine, 0.1% biocytin [Invitrogen, UK], pH 7.3, 290–310 mOsm) giving pipette resistances of 3–5 MΩ. All voltage-clamp recordings were performed at a holding potential of −65 mV and all current-clamp recordings were made from the resting membrane potential ($V_M$). In voltage-clamp mode, series resistance ($R_s$) was monitored, but not compensated. All signals were filtered online at 10 kHz using the built in 2-pole Bessel filter of the amplifiers, and digitized at 20 kHz (CED 1401, Cambridge Instruments, Cambridge, UK, NI USB-6212 BNC, National Instruments, Berkshire, UK, or Digidata 1550B, Axon Instruments, USA), using WinWCP (courtesy of John Dempster, Strathclyde University, Glasgow, UK; http://spider.science.strath.ac.uk/sipbs/software_ses.htm) or pClamp (Axon Instruments, USA). Data were analyzed offline using the open source Stimfit software package (*Guzman et al., 2014*) or MATLAB (Mathworks, USA).

We selected SOM-INs for recording on the basis of being YFP-positive cells with a soma at the *stratum* (*str.*) *oriens/alveus* border and with horizontal bipolar morphology. Recordings from CA1 PCs were obtained from YFP-negative neurons in the cell body layer. Cells were electrophysiologically characterized based on their response to a family of hyper- to depolarizing current injections (500 ms duration; −250 pA to 250 pA in 50 pA steps). Further confirmation of SOM-IN identity was based on the presence of a large voltage 'sag' in response to hyperpolarizing current steps and a non-adapting train of APs to depolarizing current. Neurons were rejected from further analysis when $V_M$ >-50 mV, if APs failed to overshoot 0 mV, initial $R_s$ exceeded 30 MΩ, or $R_s$ changed by >20% in the course of the recording.

## Characterization of presynaptic GABA_BR-mediated inhibition

Pharmacologically isolated postsynaptic currents (EPSC and IPSC) were examined in the presence of either ionotropic receptor blockers bicuculline or gabazine (10 μM, both) for EPSCs or NBQX and APV (10 and 50 μM respectively) for IPSCs, which were added to the perfusing ACSF. To evoke synaptic responses, extracellular stimuli were delivered via a glass monopolar electrode (patch pipettes filled with 2 M NaCl, pipette resistance = 0.1 MΩ) placed 50–100 μm distal from the cell body in either the alveus (EPSCs) or *str. oriens* (IPSCs). PSCs were elicited using paired stimulus (2x stimuli at 20 Hz) repeated at 0.1 Hz. Stimulus intensity was titrated to give a monosynaptic response of approximately 100 pA (range of PSC amplitudes: 19 to 366 pA). Following 5 min of stable baseline, the GABA_BR agonist R-baclofen was applied to the bath at 10 μM. In a subset of recordings, R-baclofen was added at increasing concentrations, in 5 min intervals, to assess the dose-response relationship. Following steady-state of baclofen wash-in, we removed baclofen and applied the GABA_BR antagonist CGP-55,845 (5 μM) to the bath to selectively block GABA_BRs and confirm receptor specificity. The amplitude of PSCs was measured over a 10 ms window following the stimulus artifact and mean data are presented as the average of 12 traces normalized to baseline levels over the 2 min prior to baclofen wash-in. To assess the concentration dependence of postsynaptic R-baclofen effects, recordings were made from CA1 PCs, rather than from SOM-INs, given the very low-amplitude of postsynaptic GABA_BR currents observed in these INs (*Booker et al., 2018*).

## Paired recordings from synaptically coupled CA1 PC-IN pairs

To directly assess presynaptic GABA_BR-mediated function at CA1 PC to SOM-IN synapses, we performed paired recordings between CA1 PCs and INs in *str. oriens*/alveus, as previously described (*Booker et al., 2014*). Conditions were the same as those described above, albeit with a lower intracellular EGTA concentration (0.5 mM) to prevent excessive presynaptic $Ca^{2+}$ buffering. Following characterization of intrinsic physiological responses of both pre- and postsynaptic neurons, trains of 10 APs (elicited by current pulses of 1–2 nA, 1 ms, 20 Hz) were delivered to the PC while recording the IN in voltage-clamp at −65 mV. A unitary synaptic connection was confirmed as a short latency (<4 ms) EPSCs following the presynaptic APs detected in averages of 10 traces. If synaptic connectivity was not observed in the IN, the recording was abandoned, an outside-out patch formed, and a neighboring CA1 PC recorded. Once a synaptic connection was found, we recorded >50 traces with unitary EPSCs from the interneurons elicited by APs evoked in the PCs by brief current pulses (1–2 nA, 1 ms) every 5 s, then applied 2 μM R-baclofen to the bath for 5 min, followed by application of CGP-55,845 (5 μM), without baclofen. The EPSC amplitude was measured from the preceding baseline as an average over a 0.4 ms window corresponding to the peak region of the synaptic responses within 10 ms from the start of the AP. Mean unitary EPSCs are shown and measured from at least 30 traces.

## Optogenetic activation of SOM-INs

Whole-cell recordings were obtained from CA1 PCs in acute slices (300 μm thick) held at −70 mV in acute slices from mice expressing ChR2 and tdTom specifically in SOM-INs. IPSCs were evoked by pulses of blue light (473 nm; 2 ms; CoolLED system, UK) centered at the border of *str. radiatum* and *lacunosum-moleculare* repeated at 5 s intervals. Basal synaptic transmission was measured and the effect of R-baclofen (10 μM) was analyzed after 5 min of bath application. In a subset of experiments (4 out of 6), CGP 55845 (5 μM), without baclofen, was subsequently bath applied and the prior effect of baclofen was fully reversed. In a subset of recordings, 10 μM SCH-23,390 was preapplied to the bath, after which first baclofen and then CGP were co-applied with SCH-23,390. Data were obtained from averages of 30–50 traces except for the figure showing the time course of drug effects.

## Generation of network oscillations and field potential recordings

To preserve a larger intact local network, thicker, 400 μm acute hippocampal slices were prepared (as above) in either the transverse (gamma oscillations) or longitudinal plane (theta oscillations) (*Gloveli et al., 2005a*), then stored in a liquid/gas interface chamber. For theta oscillations, slices were moved from the interface chamber, into a submerged chamber perfused with ACSF containing 50 μM carbachol at a rate of 10–12 ml/minute. An extracellular recording electrode made of a patch pipette filled with ACSF was carefully placed in proximal *str. radiatum* and the field response

recorded. For gamma oscillations slices were transferred to the submerged recording chamber and a pressure application ('puff') pipette containing 2 mM kainate was placed in the distal *str. radiatum.* Puffs of kainate (10 psi, 20 ms, repeated at 1 min intervals) were applied to the slice, which invariably resulted in gamma oscillations in the field. Once a stable oscillation was confirmed a SOM-IN was recorded from *str. oriens* and a minimum of 5 min of theta or 5 kainate puffs were collected following recovery of the oscillation to the previous state. Given the $EC_{50}$ of R-baclofen being ~1 µM at presynaptic $GABA_BRs$, we applied 2 µM R-baclofen to the bath and following wash in (2 mins) a further 5 min of oscillation data were collected. In a subset of recordings, to facilitate AP discharge, a depolarizing bias current 10 pA below rheobase was applied to the recorded neuron. In a further subset of recordings a second puff pipette filled with 2 mM baclofen dissolved in 150 mM NaCl was placed in *str. oriens* proximal to the recorded IN (<100 µm). First 3 kainate puffs alone were applied to collect control data, then 3 puffs where baclofen puff preceded the kainate puff by 100 ms were applied to examine the effects of $GABA_BR$ activation. In some recordings CGP (5 µM) was applied following R-baclofen to confirm the receptor specificity.

Prior to analysis, all field recordings were band-pass filtered using a Butterworth-filter at 4–20 Hz (Theta) or 30–200 Hz (Gamma). Peak oscillatory power was measured using fast-Fourier transform (FFT)-based spectral analysis (Spike2 software, CED, Cambridge, UK). Peak frequency and power were measured across 5 min of theta activity following carbachol wash in or during the initial 10 s of gamma activity evoked by the kainate puff. Spike triggered averages were produced using a custom MATLAB script, over either a 150 ms (Theta) or 25 ms window (Gamma) (code available on GitHub: https://github.com/imrevida/eLife-Booker-2020-MatlabCode; *Vida, 2020*; copy archived at https://github.com/elifesciences-publications/eLife-Booker-2020-MatlabCode). The relative phase of each AP was determined according to the Hilbert transform and plotted with respect to the full cycle of each oscillation. Mean AP frequency was measured as the number of spikes observed over the respective recording window.

## Visualization, imaging and reconstruction of the recorded neurons

Immunocytochemistry was performed to identify recorded neurons (*Booker et al., 2014*). Following experiments, slices were fixed in 4% paraformaldehyde diluted in 0.1 M PB overnight (O/N) at 4°C. Slices were rinsed in PB, then phosphate-buffered saline (PBS; 0.025 M PB and 0.9% NaCl) and blocked with 10% normal goat serum (NGS) with 0.3–0.5% TritonX-100 and 0.05% $NaN_3$ diluted in PBS for 1 hr at RT. Slices were then incubated for 48–72 hr in a PBS solution containing 5% NGS, 0.3–0.5% TritonX-100 and 0.05% $NaN_3$ and primary antibodies against SOM (rabbit polyclonal, 1:2000, Peninsula Laboratories, USA), at 4°C. Slices were subsequently rinsed extensively in PBS for an hour and then incubated with fluorescently conjugated secondary antibodies raised against rabbit (Goat-anti rabbit AlexaFluor 405, 488 or 546; 1:500, Invitrogen, Dunfermline, UK) as well as fluorescently conjugated streptavidin (AlexaFluor 647; 1:500, Invitrogen) in a PBS solution containing 3% NGS, 0.1% TritonX-100% and 0.05% $NaN_3$ O/N at 4°C. Slices were rinsed in PBS, then PB, and mounted on glass slides with a polymerizing mounting medium (Fluoromount-G, Southern Biotech, AL, USA) and cover-slipped. Filled neurons were imaged with a laser scanning confocal microscope (FluoView 1000, Olympus) under a 20x (N.A 0.75) objective and z-axis stacks of images (2048 × 2048 pixels, at 1 µm axial steps) collected to allow identification of somatodendritic and axonal arborizations. To assess immunoreactivity of the recorded cells the somata and proximal dendrites of neurons were imaged with a silicon-immersion 60x (N.A. 1.3) objective lens and either single confocal images or stacks of 5–10 images at 1 µm axial steps were taken. Selected, representative cells were reconstructed off-line from 20x magnification image stacks digitally stitched using semi-automatic analysis software (Simple Neurite Tracer plug-in for the FIJI software package, http://fiji.sc) (*Longair et al., 2011*).

## Sodium dodecyl sulfate-digested freeze-fracture replica immunolabeling (SDS-FRL)

To determine the distribution pattern and density of $GABA_BRs$ on presynaptic boutons contacting SOM-INs and axon terminals of the INs, the $GABA_{B1}$ subunit was detected with SDS-FRL as previously described (*Booker and Vida, 2018*). Male mice (6-week-old, n = 3) in which SOM-INs selectively expressed ChR2-YFP fusion protein were derived from crossing SOM-Cre and Ai32 transgenic

mice. These mice and male Wistar rats (8-week-old, n = 2) were sedated with isoflurane and then terminally anesthetized with pentobarbital (80 mg/kg for mice and 50 mg/kg for rats, intraperitoneally). Animals were then transcardially perfused with 0.9% NaCl for 1 min followed by a fixative containing 1% paraformaldehyde and 15% saturated picric acid in 0.1 M PB (pH 7.4) for 12 min. Hippocampal slices (130 µm) were cut on a vibratome (VT 1000, Leica, Vienna, Austria) and cryoprotected with 30% glycerol in 0.1 M PB O/N at 4°C. Blocks containing all layers of the CA1 area were trimmed from the slices and frozen under high-pressure (HPM 100, Leica). The frozen samples were fractured at −140°C and the fractured facets were coated with carbon (5 nm), platinum-carbon (2 nm) and an additional layer of carbon (18 nm) in a freeze-fracture replica machines (ACE 900, Leica or BAF 060, BAL-TEC, Lichtenstein). Replicas were digested at 60°C in a solution containing 2.5% SDS and 20% sucrose diluted in 15 mM Tris buffer (TB, pH 8.3) for 48 hr followed by 37°C for 18 hr, washed in washing buffer comprising 0.05% bovine serum albumin (BSA, Roth, Germany) and 0.1% Tween 20 in 50 mM Tris-buffered saline (TBS) and then blocked in a solution containing 5% BSA and 0.1% Tween 20 in TBS for 1 hr at RT. Afterwards, replicas were incubated at 15°C for 2 days in the following mixtures of primary antibodies in a solution containing 1% BSA and 0.1% Tween 20 made up in TBS: (i) $GABA_{B1}$ (B17, rabbit, 10 µg/ml; *Kulik et al., 2002*) or A25, rabbit, 5 µg/ml; *Engle et al., 2006*), green fluorescence protein (GFP-1010, chicken, 0.4 µg/ml, Aves Labs, Oregon) and $Ca_v2.1$ (Guinea pig, 4 µg/ml, Frontier Institute, Hokkaido, Japan; *Althof et al., 2015*) or (ii) $GABA_{B1}$, GFP and vesicular glutamate transporter 1 (VGluT1, goat, 0.5 µg/ml, Frontier Institute, Hokkaido; *Kusch et al., 2018*) or (iii) $GABA_{B1}$, GFP and vesicular GABA transporter (VGAT, Guinea pig, 4.5 µg/ml, Frontier Institute, Hokkaido; *Althof et al., 2015*) or (iv) $GABA_{B1}$, GFP, VGluT1 and metabotropic glutamate receptor 1α-subunit (mGluR1α, Guinea pig, 0.4 µg/ml, Frontier Institute, Hokkaido; *Booker et al., 2018*). Replicas were washed in washing buffer then reacted with 6 nm, 12 nm and 18 nm gold particle-conjugated secondary antibodies (1:30, Jackson ImmunoResearch Europe, Cambridgeshire) or with the aforementioned antibodies together with 3 nm gold particle-conjugated secondary antibody (1:800, Nanopartz, Colorado) O/N at 15°C. Finally, replicas were washed in TBS then distilled water and mounted on Formvar-coated 100-mesh grids.

## Electron microscopy

Replicas were analyzed with an electron microscope (Zeiss Leo 912 omega, Carl Zeiss, Oberkochen, Germany). All antibodies target intracellular epitopes of proteins, therefore, immunoreactivity can be observed on the protoplasmic face (P-face) of the plasma membrane. Boutons contacting YFP-positive dendritic shafts of INs in *str. oriens-alveus* and YFP-positive axon terminals of INs in *str. lacunosum-moleculare* were sampled. Densely spiny CA1 PC dendrites in *str. radiatum*, YFP-positive dendritic shafts of SOM-INs in *str. oriens-alveus*, and YFP-negative boutons of putative PCs in *str. oriens* were used as a control for $GABA_{B1}$ labeling.

## Chemicals and pharmacology

All chemicals were obtained from either Sigma Aldrich (Munich, Germany) or Carl Roth (Karlsruhe, Germany). Biocytin was obtained from Life Technologies (Dunfermline, UK). Pharmacological agents were obtained from Abcam Biochemicals (Cambridge, UK) or Tocris Bioscience (Bristol, UK). Drugs were stored as 1000-fold concentrated stocks at −80°C until used. Working solutions were prepared fresh on the day in normal ACSF at final concentrations given in the text.

## Statistical analysis

Statistical analysis was performed with Graphpad Prism (GraphPad Software, CAUSA). Analysis of unpaired data was performed with Mann-Whitney U tests. Group data were compared with one-way ANOVA tests, combined with Holm-Sidak post-tests. Paired group data were analyzed with one-way repeated measures ANOVA. Data are shown as mean ± SEM throughout. Statistical significance was assumed if $p < 0.05$.

## Acknowledgements

We thank Natalie Wernet and Ina Wolter for excellent technical support. Funding was provided by: DFG (FOR 2134, AK, MB, IV), BIOSS-2 (AK), McNaught Bequest (IV, SAB), and Tenovus Scotland (IV). $GABA_{B1}$ (A25) antibody was generously provided by Dr. Bernhard Bettler (*Engle et al., 2006*). We

also acknowledge support from the German Research Foundation (DFG) and the Open Access Publication Funds of Charité – Universitätsmedizin Berlin.

## Additional information

### Competing interests

Marlene Bartos: Reviewing editor, *eLife*. The other authors declare that no competing interests exist.

### Funding

| Funder | Grant reference number | Author |
|---|---|---|
| Deutsche Forschungsgemeinschaft | FOR 2134 | Marlene Bartos<br>Akos Kulik<br>Imre Vida |
| Deutsche Forschungsgemeinschaft | BIOSS-2 | Akos Kulik |
| Tenovus | PROJECT S07/6 | Imre Vida |
| University of Glasgow | McNaught Bequest | Sam A Booker<br>Imre Vida |
| Charité – Universitätsmedizin Berlin | Open Access Publication Funds | Imre Vida |

The funders had no role in study design, data collection and interpretation, or the decision to submit the work for publication.

### Author contributions

Sam A Booker, Conceptualization, Formal analysis, Investigation, Visualization; Harumi Harada, Formal analysis, Investigation, Visualization; Claudio Elgueta, Julia Bank, Formal analysis, Investigation; Marlene Bartos, Conceptualization, Resources, Supervision; Akos Kulik, Conceptualization, Formal analysis, Supervision, Funding acquisition; Imre Vida, Conceptualization, Resources, Supervision, Funding acquisition

### Author ORCIDs

Sam A Booker https://orcid.org/0000-0003-1980-9873
Harumi Harada https://orcid.org/0000-0001-7429-7896
Marlene Bartos http://orcid.org/0000-0001-9741-1946
Imre Vida https://orcid.org/0000-0003-3214-2233

### Ethics

Animal experimentation: Care and handling of the animals prior to and during the experimental procedures followed European Union and national regulations (German Animal Welfare Act; ASPA, United Kingdom Home Office) and all experiments were performed in accordance with institutional guidelines (Charité- Universitätmedizin Berlin; University of Freiburg, Freiburg, Germany), with permissions from local authorities (LaGeSo, Berlin, T-0215/11 LaGeSo; Freiburg, X14/11H and 35-9185.81/G-19/59).

### Decision letter and Author response

Decision letter https://doi.org/10.7554/eLife.51156.sa1
Author response https://doi.org/10.7554/eLife.51156.sa2

## Additional files

### Supplementary files

• Transparent reporting form

## Data availability

Quantitative electrophysiological, optogenetic and immuno-electron microscopic data presented in the figures and text has been deposited to Dryad (https://doi.org/10.5061/dryad.gt160v2).

The following dataset was generated:

| Author(s) | Year | Dataset title | Dataset URL | Database and Identifier |
|---|---|---|---|---|
| Booker SA, Harada H, Elgueta C, Bank J, Bartos M, Kulik A, Vida I | 2020 | Data from: Presynaptic GABAB receptors functionally uncouple somatostatin interneurons from the active hippocampal network | https://doi.org/10.5061/dryad.gt160v2 | Dryad Digital Repository, 10.5061/dryad.gt160v2 |

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
