## [Decision Letter]

**Acceptance summary:**

The findings of this study significantly advance our understanding of the modulation of synaptic activity in the context of hippocampal network function. Using freeze-fracture replica immunolabelling, the authors provide high quality anatomical support for modulation of somatostatin interneurons by GABA_B_ receptors, and combine this with whole cell electrophysiological recordings and optogenetics to probe the role of these receptors in hippocampal function in vitro.

**Decision letter after peer review:**

Thank you for submitting your article "Presynaptic GABA_B_ receptors functionally uncouple somatostatin interneurons from the active hippocampal network" for consideration by *eLife*. Your article has been reviewed by three peer reviewers, and the evaluation has been overseen by a Reviewing Editor and Laura Colgin as the Senior Editor. The following individuals involved in review of your submission have agreed to reveal their identity: Francesco Ferragutti (Reviewer #2).

The reviewers have discussed the reviews with one another and the Reviewing Editor has drafted this decision to help you prepare a revised submission.

Summary:

The authors of this manuscript have investigated whether the activation of presynaptic GABA_B_ receptors can modulate the synaptic inputs onto the SOM+ hippocampal interneurons as well as their output synapses. Using pharmacological tools the authors unequivocally demonstrated that GABA_B_ receptors are in a position to critically influence the transmitter release at both the input and output synapses of SOM+ interneurons. The findings of this study are of high scientific interest as they significantly advance our understanding of the modulation of synaptic activity by GABA_B_ receptors at the SOM interneuron feedback hippocampal microcircuit. Most of the experiments described in this paper are very well performed. Both the anatomical localization of GABA_B_ receptors using the freeze-fracture replica immunolabelling approach, a technique mastered by the Kulik's lab, and the functional characterization of the influence of GABA_B_ receptors on the SOM interneuron feedback microcircuit by means of whole-cell recordings and in vitro optogenetics are of very high quality. In the last part of the paper, the authors provide indirect evidence for a network effect that is dependent on GABA_B_ receptors influence on SOM+ cells: they propose that specific activation of presynaptic GABA_B_ receptors could have an impact on the coupling of SOM+ interneuron spiking during network oscillations induced in slice preparations. However, result show that with “global” GABA_B_ activation SOM+ IN spiking was substantially reduced in a way that was correlated with the decrease in oscillation power. This last finding regarding specific network effects needs stronger experimental support.

Essential revisions:

1) Bath application of 2 µM baclofen caused a marked reduction in the peak amplitude of network oscillations as well as in the firing rate of OLM+ neurons. Based on the present findings, GABA_B_ receptors can reduce the ESPCs onto SOM+ interneurons, leading to de-coupling of their firing from the ongoing oscillations, as proposed by the Authors. However, based on these results it is hard to conclude that presynaptic GABA_B_ receptors are mainly responsible for de-coupling of SOM+ IN activity from the oscillations. Many in vitro and in vivo studies have shown that the IN firing is primarily driven by phasic EPSCs (i.e. synchronous discharge of pyramidal cells) during oscillatory activities. Therefore, GABA_B_ receptor activation, impacting both the fast spiking interneuron and the pyramidal cell operation (Booker et al., 2013, 2017), can first cause a reduction in the oscillation power by altering the oscillogenesis, which leads to the de-synchronization of EPSCs, and therefore to the suppression of phase-coupled spiking in any interneuron types. Using the applied methods, comparing the features of spiking and oscillation in three discrete time points, it is hard to distinguish between these two scenarios, which may actually work in parallel. This tentative result requires additional experimental evidence. One experimental approach that might provide more direct evidence for the decoupling hypothesis (we are sure the authors will have many other ideas) is the following: Local application of baclofen into the surrounding of the SOM IN while spiking is monitored. This approach can minimize the effect of the baclofen at the postsynaptic GABA_B_ receptors on the pyramidal cells, i.e. the oscillogenesis will not be impaired. Yet, synchronous EPSCs from the PCs could be reduced by locally applied baclofen via presynaptic GABA_B_ receptors, leading to a reduction in EPSC magnitude. If this effect is large enough, then activation of presynpatic GABA_B_ receptors could suppress the phasic excitation received by SOM INs without affecting the gamma oscillation globally. Under these circumstances, SOM IN spiking should be uncoupled from the on-going oscillation, supporting the original claim.

Overall, the authors conclusions on this point do not adequately consider that in their experimental paradigm the activation of presynaptic GABA_B_ receptors on glutamatergic and GABAergic terminals innervating SOM interneurons and on terminals from these interneurons onto pyramidal cells occurred simultaneously, a situation highly unlikely in vivo.

Further, activation of GABA_B_ receptors at presynaptic GABAergic terminals (presumably originating from VIP/CR interneurons) onto SOM interneurons should: i) reduce quite effectively the release of GABA (also in view of their high density at these terminals) from VIP/CR interneurons and consequently also the spillover onto glutamatergic synapses, hence, lifting, at least in part, the heterosynaptic inhibition onto feedback inputs from pyramidal cells; ii) lead to disinhibition of SOM interneurons; iii) increase their gating of entorhinal inputs. This would be quite different from a functional uncoupling.

2) SOM+ interneurons in the CA1 hippocampus are diverse as they belong to at least two categories: OLM cells and GABAergic projecting neurons, often innervating the medial septum. Would it be possible to split at least a portion of the recorded interneurons into the two groups and compare whether the inputs of these two interneuron types are controlled similarly or distinctly by presynaptic GABA_B_ receptors? The categorization can be based on the morphological features, including the axon arborization, the morphology of the dendrites, the location of the axon initial segments (axon initiates usually from the dendrites in case of OLM cells, whereas the axon initial segment of projecting neurons originates from the soma), and/or the single cell properties such as the 'sag' characteristics (Zemankovics et al., 2010). Comparison performed even on a subset of interneurons may provide some valuable insights how the inputs onto these functionally distinct inhibitory cells are controlled via GABA_B_ receptors.

3) The authors claim a different sensitivity (EC50) between presynaptic (1.1 µM) and postsynaptic (4.3 µM) GABA_B_ receptors to baclofen. However, based on the confidence intervals deduced from the dose response curves shown in Figure 1D, I would conclude that there is no difference in the EC50 between pre- and post-synaptic GABA_B_ receptors. The claim that a pharmacological difference exists between these two GABA_B_ receptor pools must be removed, and, likewise, any statement that a certain concentration of baclofen (e.g. 2 µM) would preferentially activates one of the pools.

4) In subsection “GABA_B_R Subunits are Expressed at Presynaptic Boutons Forming Synapses Onto SOM-INs”, it is mentioned that the density of GABA_B1_ labelling is higher in VGluT1-negative boutons in comparison with VGluT1-positive boutons. Was it quantified? If so, how did it compare with the data shown in Figure 4D?

5) The manuscript needs additional editing in order to fully understand the text. There are many sentences that are difficult to understand, and there are some apparent grammatical/syntactic errors.

a) Supplementary Figure 1. Related to Figure 1 has a wrong legend.

There are two figures called: "Supplementary Figure 1" and "Supplementary Figure 1. Related to Figure 1". They have the same legend but they have different images.

For his reason it is very difficult to read the section about EM.

– Paragraph two of subsection “GABA_B_R Subunits are Expressed at Presynaptic Boutons Forming Synapses Onto SOM-INs”: (data not shown), please show this data.

b) Statistical analysis: In the methods, Mann-Whitney or Wilcoxon tests were used. This contrasts with the results, in which it never seems to be mentioned Wilcoxon test was used. This is especially important for the data in Figure 1F for PPR and later for experiments with SCH. It is incorrect to say that "… lower than the control IPSCs in the group without SCH application, albeit not significantly different….". Mann-Whitney test was used to check statistics. However, this is a paired recording and you should use e.g. Wilcoxon test.

c) Does the application of Kir3 channel blocker influence IPSC amplitude, input resistance, and/or whole-cell currents in PC and SOM? If Kir3 blocker affects these parameters maybe you should consider reverse experiments (SCH after baclofen)? Would you see the same effect as during experiments with baclofen after SCH? Do you mean by your experiments that baclofen does not activate postsynaptic GABA_B_ receptors onto PC cells?

d) What is the effect of baclofen on the whole-cell current in SOM neurons and PC cells?

6) Major concerns for methods:

– The rationale for two different approaches to incorporate ChR2-YFP proteins into SOM INs is not clear.

What was the reason for virus injections vs. Ai32 mice? Were there any differences?

Subsection “Acute slice preparation”: "…slices (300 or 500)…", but in the legend of Figure 7 indicates 400.

– Subsection “Characterization of presynaptic GABA_B_R-mediated inhibition”: The response was about 100 pA (range of intensities: 19-366 pA). It is unclear what "intensities" is related to (the amplitude or the stimulus intensity?). How is the amplitude related to the maximal response? PPR is strongly dependent on the stimulus strength; the effect of baclofen might be also dependent on the stimulation intensity. Could you comment this?

– Subsection “Paired recordings from synaptically coupled CA1 PC-IN pairs”: Could you explain why you use two different concentration of EGTA: 10 and then 0.5 mM?

7) Discussion:

Several sentences are unclear. Please revise for lucidity: Paragraph one and two: What is "them" related to? SOM-Ins or PCs or both?

– Figure 5: E: it is unclear how many cells are there in fact. If points are overlaid, please arrange them in the way you use in Figure 4.

– Figure 7: Consider different, more distinct colors, clarify legend.

---

## [Author Response]

Essential revisions:1) Bath application of 2 µM baclofen caused a marked reduction in the peak amplitude of network oscillations as well as in the firing rate of OLM+ neurons. Based on the present findings, GABA_B_ receptors can reduce the ESPCs onto SOM+ interneurons, leading to de-coupling of their firing from the ongoing oscillations, as proposed by the Authors. However, based on these results it is hard to conclude that presynaptic GABA_B_ receptors are mainly responsible for de-coupling of SOM+ IN activity from the oscillations. Many in vitro and in vivo studies have shown that the IN firing is primarily driven by phasic EPSCs (i.e. synchronous discharge of pyramidal cells) during oscillatory activities. Therefore, GABA_B_ receptor activation, impacting both the fast spiking interneuron and the pyramidal cell operation (Booker et al., 2013, 2017), can first cause a reduction in the oscillation power by altering the oscillogenesis, which leads to the de-synchronization of EPSCs, and therefore to the suppression of phase-coupled spiking in any interneuron types. Using the applied methods, comparing the features of spiking and oscillation in three discrete time points, it is hard to distinguish between these two scenarios, which may actually work in parallel. This tentative result requires additional experimental evidence. One experimental approach that might provide more direct evidence for the decoupling hypothesis (we are sure the authors will have many other ideas) is the following: Local application of baclofen into the surrounding of the SOM IN while spiking is monitored. This approach can minimize the effect of the baclofen at the postsynaptic GABA_B_ receptors on the pyramidal cells, i.e. the oscillogenesis will not be impaired. Yet, synchronous EPSCs from the PCs could be reduced by locally applied baclofen via presynaptic GABA_B_ receptors, leading to a reduction in EPSC magnitude. If this effect is large enough, then activation of presynpatic GABA_B_ receptors could suppress the phasic excitation received by SOM INs without affecting the γ oscillation globally. Under these circumstances, SOM IN spiking should be uncoupled from the on-going oscillation, supporting the original claim.

We agree that bath application of baclofen likely attenuates presynaptic release of both GABA and glutamate in the wider network, plausibly explaining the reduction in gamma and theta oscillation power observed over the spectra and peak power plots (Figure 7B and F). To address these concerns, we followed the suggestion of the reviewers and performed dedicated experiments in which we focally applied baclofen in the proximity of a recorded SOM-IN in *str. oriens* (<100 µm from the IN somata) during kainate-induced gamma oscillations (Results and new Figure 7—figure supplement 2). These data show that there is no change to the gamma power induced by kainate application to *str. radiatum* when baclofen is focally applied. Nevertheless, focal baclofen application produces a strong reduction in SOM-IN AP discharge, supporting our hypothesis of an uncoupling of the INs from the network by presynaptic GABA_B_R activation.

We performed the new experiments in mouse slices, whereby we could also show that the strong presynaptic inhibition of EPSCs onto SOM-INs is not species specific (Results and new Figure 1—figure supplement 1), and that gamma power, as well as, the discharge of the INs are strongly attenuated by bath application of baclofen in mice, too (Results and new Figure 7—figure supplement 1).

Together these new data show that local activation of GABA_B_ receptors at axon terminals onto SOM-INs during ongoing gamma oscillations is sufficient to uncouple and silence this IN population and that this property is shared between the two rodent species.

Overall, the authors conclusions on this point do not adequately consider that in their experimental paradigm the activation of presynaptic GABA_B_ receptors on glutamatergic and GABAergic terminals innervating SOM interneurons and on terminals from these interneurons onto pyramidal cells occurred simultaneously, a situation highly unlikely in vivo.

We agree with the reviewer that GABA activating of GABA_B_Rs in the *str oriens*, where SOM-INs soma and dendrites and their input synapses are localized, *versus* the str. l-m., where their axons terminate, is likely to originate from distinct sources and therefore occur independently. Nevertheless, network mechanisms may coordinate these sources and extracellular GABA levels can surge simultaneously in the two layers under certain conditions.

We have added a sentence to the Discussion to raise this point.

Further, activation of GABA_B_ receptors at presynaptic GABAergic terminals (presumably originating from VIP/CR interneurons) onto SOM interneurons should: i) reduce quite effectively the release of GABA (also in view of their high density at these terminals) from VIP/CR interneurons and consequently also the spillover onto glutamatergic synapses, hence, lifting, at least in part, the heterosynaptic inhibition onto feedback inputs from pyramidal cells; ii) lead to disinhibition of SOM interneurons; iii) increase their gating of entorhinal inputs. This would be quite different from a functional uncoupling.

This is indeed a very interesting point; however, we don’t see a way to selectively activate presynaptic receptors at GABAergic input synapses onto SOM-INs. One pertinent point is that it is unclear what are the main sources of GABA for the activation of presynaptic GABA_B_R at these synapses. One possibility is that GABA is released from VIP/CR INs, as the reviewer suggests. Alternatively, other IN types, most notably Ivy cells, but also Bistratified cells or even SOM-INs themselves, could be the source of GABA in this layer. In our experimental paradigm (including the newly performed experiments suggested by the reviewer), we assumed that the increase in extracellular GABA concentration occurs due to spillover (mimicked by baclofen application) and thereby produces heterosynaptic inhibition. Under these conditions reduced GABAergic transmission onto SOM-INs matches the reduction of glutamatergic transmission in line with a network uncoupling, rather than leading to disinhibition of the INs.

2) SOM+ interneurons in the CA1 hippocampus are diverse as they belong to at least two categories: OLM cells and GABAergic projecting neurons, often innervating the medial septum. Would it be possible to split at least a portion of the recorded interneurons into the two groups and compare whether the inputs of these two interneuron types are controlled similarly or distinctly by presynaptic GABA_B_ receptors? The categorization can be based on the morphological features, including the axon arborization, the morphology of the dendrites, the location of the axon initial segments (axon initiates usually from the dendrites in case of OLM cells, whereas the axon initial segment of projecting neurons originates from the soma), and/or the single cell properties such as the 'sag' characteristics (Zemankovics et al., 2010). Comparison performed even on a subset of interneurons may provide some valuable insights how the inputs onto these functionally distinct inhibitory cells are controlled via GABA_B_ receptors.

We have examined the morphology of the recorded and immunocytochemically-identified SOM-INs. Of a total of 63 rat SOM-INs, 25 cells possessed an axon projecting to the *str. lacunosum-moleculare,* 2 INs had bistratified axons in *str. radiatum* and *oriens*, and 4 had axons confined to *str. oriens* only. The remaining INs either had axons cut close to the soma (8 cells) or were not sufficiently filled to allow morphological identification (24 cells). Of the 8 cells with cut axon, 7 had the axon clearly emerging from a primary dendrite, and only 1 had an axon originating from the soma. As such, we believe that our data reflect an enriched population of OLM cells. We have now included the above morphological data in the Results text.

3) The authors claim a different sensitivity (EC50) between presynaptic (1.1 µM) and postsynaptic (4.3 µM) GABA_B_ receptors to baclofen. However, based on the confidence intervals deduced from the dose response curves shown in Figure 1D, I would conclude that there is no difference in the EC50 between pre- and post-synaptic GABA_B_ receptors. The claim that a pharmacological difference exists between these two GABA_B_ receptor pools must be removed, and, likewise, any statement that a certain concentration of baclofen (e.g. 2 µM) would preferentially activates one of the pools.

We appreciate the reviewers concerns and have removed the data concerning pharmacological differences from Figure 1 and from the Results.

4) In subsection “GABA_B_R Subunits are Expressed at Presynaptic Boutons Forming Synapses Onto SOM-INs”, it is mentioned that the density of GABA_B1_ labelling is higher in VGluT1-negative boutons in comparison with VGluT1-positive boutons. Was it quantified? If so, how did it compare with the data shown in Figure 4D?

The immunoparticle density for GABA_B1_ was initially quantified in VGluT1+/VGAT- excitatory and VGluT1-/VGAT+ inhibitory terminals obtained from mouse brains, because dendrites of SOM-INs selectively expressed ChR2-YFP fusion protein, which enabled us to analyze boutons making synapses on identified dendritic shafts of neurons of interest. The surface density of the receptor subunit in rat samples has also been determined, but using another GABA_B1_ subunit-specific antibody (A25), due to the limited availability of B17 antibody. Because the intensity of immunolabelling was generally weaker with A25 than with B17, the absolute values cannot be directly compared. However, the relative density of GABA_B1_ in the two groups of terminals was the very similar: mouse inhibitory terminals possess 2.5-fold higher mean density of GABA_B1_ than excitatory boutons (43.00 particles/µm^2^*vs*. 17.37 particles/µm^2^, p=0.0001, Mann-Whitney test; see also Results), whereas rat inhibitory boutons show 2.7-fold higher mean density for the subunit than excitatory terminals (30.03 particles/µm^2^*vs*. 11.00 particles/µm^2^, p=0.0002, Mann-Whitney U test). Please also see the bar graphs in Author response image 1.

**Author response image 1. respfig1:** Summary bar charts of the surface densities of immunogold particles for GABA_B1_ subunit in excitatory and inhibitory axon terminals forming synapses onto SOM-IN dendrites in the mouse (left) and the rat hippocampus (right) detected using the B17 and A25 primary antibodies, respectively (see Materials and methods for further detail). Superimposed open circles represent density values from individual boutons.

5) The manuscript needs additional editing in order to fully understand the text. There are many sentences that are difficult to understand, and there are some apparent grammatical/syntactic errors.

We apologize for typographical and syntactic errors. We have addressed all highlighted problems and made effort to eliminate any further mistakes in the revised version.

a) Supplementary Figure 1. Related to Figure 1 has a wrong legend.There are two figures called: "Supplementary Figure 1" and "Supplementary Figure 1. Related to Figure 1". They have the same legend but they have different images.For his reason it is very difficult to read the section about EM.

We thank the reviewer flagging this up and apologize for the ambiguity. This has been corrected in the revised version. Please note that due to the inclusion of new figures the supplemental figures were also renumbered.

– Paragraph two of subsection “GABA_B_R Subunits are Expressed at Presynaptic Boutons Forming Synapses Onto SOM-INs”: (data not shown), please show this data.

We have now included representative images in a new Figure 3—figure supplement 2 to illustrate the statement made in the Results text.

b) Statistical analysis: In the methods, Mann-Whitney or Wilcoxon tests were used. This contrasts with the results, in which it never seems to be mentioned Wilcoxon test was used. This is especially important for the data in Figure 1F for PPR and later for experiments with SCH. It is incorrect to say that "… lower than the control IPSCs in the group without SCH application, albeit not significantly different….". Mann-Whitney test was used to check statistics. However, this is a paired recording and you should use e.g. Wilcoxon test.

We have thoroughly checked and revised the Materials and methods and the Results to reflect the tests employed. We, in fact, did not use the Wilcoxon test, as the final data was better reflected by a 1-way repeated-measures ANOVA to account for within cell effects – as in Figure 1D-G. For post-hoc tests, we uniformly used the Holm-Sidak test, adjusted for multiple comparisons.

With regard to the specific example the comparison made here is between the IPSC recorded under control conditions in one group of cells and the IPSC elicited in the presence of SCH in another group of cells. As such this is not a paired analysis, and for this non-normally distributed data we believe the Mann-Whitney test is more appropriate.

c) Does the application of Kir3 channel blocker influence IPSC amplitude, input resistance, and/or whole-cell currents in PC and SOM? If Kir3 blocker affects these parameters maybe you should consider reverse experiments (SCH after baclofen)? Would you see the same effect as during experiments with baclofen after SCH? Do you mean by your experiments that baclofen does not activate postsynaptic GABA_B_ receptors onto PC cells?

The aim of applying of SCH was to eliminate potential shunting effects due to postsynaptic GABA_B_ receptor activation mediated by Kir3 channels. As we had stated in the Results before, SCH application blocks a large portion of the baclofen-induced postsynaptic Kir3 current in CA1 PCs. Nevertheless, the reduction in the amplitude of the IPSCs in response to baclofen application in the presence and absence of SCH was comparable, indicating that the reduced transmission was primarily reflecting a presynaptic inhibitory effect.

To address the issue raised by the reviewer, we have included an additional figure (new Figure 5—figure supplement 1) summarizing the effects of baclofen, CGP and SCH on whole-cell currents in CA1 PCs. Consistent with the lack of tonic postsynaptic GABA_B_-mediatedcurrents in PCs (new Figure 5—figure supplement 1B; see also Degro et al., 2015), we haven’t noticed any effect of SCH on whole-cell current or input resistance in PCs. Finally, IPSCs amplitudes, even if they tended to be smaller in our sample of cells recorded in the presence of SCH, were not significantly different from the control sample (see Figure 5C and D, and the relevant section in the Results).

Regarding the effects in SOM-INs, as we have shown previously, GABA_B_Rs on SOM-IN dendrites only minimally activate Kir3 channels (Booker et al., 2018); therefore we would not expect any large impact of SCH application in these cells in the absence or presence of baclofen. A thorough characterization of the postsynaptic interactions of baclofen and SCH in PCs and SON-INs, in our view, would be beyond the scope of the present study.

d) What is the effect of baclofen on the whole-cell current in SOM neurons and PC cells?

We addressed this question in our previous study (Booker et al., 2018), in which we showed that pharmacological activation of GABA_B_Rs in the absence of GABAergic and glutamatergic synaptic inputs (DL-AP5, CNQX, and Bicuculline) leads to minimal post-synaptic currents. A comment to this effect is included now at the beginning of the Results. Summary data on whole-cell currents measured in baclofen, SCH and CGP in CA1 PCs has been added as a new Figure 5—figure supplement 1 to the revised version.

6) Major concerns for methods:– The rationale for two different approaches to incorporate ChR2-YFP proteins into SOM INs is not clear.What was the reason for virus injections vs. Ai32 mice? Were there any differences?

The physiological and anatomical investigations were not simultaneously performed and at the time of the optogenetic experiments Ai32 animals were not yet available for breeding.

However the distinct approaches seemed to be functionally favourable for the experiments as the viral injection enabled a focal transfection of SOM-INs and supported photostimulation of a lower number of local axons synapsing onto the recorded CA1 PCs, whereas the SOM-Cre x Ai32 animals facilitated an efficient SDS-FRIL-EM characterization of SOM-INs input and output synapses.

Subsection “Acute slice preparation”: "…slices (300 or 500)…", but in the legend of Figure 7 indicates 400.

Thank you for pointing out this inconsistency. This should have read 400 µm for all in vitro oscillation experiments, as we used slightly thicker slices to preserve a larger intact local network. We used standard 300 µm acute slices for all other experiments as these provide better optics for cell identification, whilst keeping most cellular processes intact. This ambiguity has been corrected at all the relevant locations in the Materials and methods.

– Subsection “Characterization of presynaptic GABA_B_R-mediated inhibition”: The response was about 100 pA (range of intensities: 19-366 pA). It is unclear what "intensities" is related to (the amplitude or the stimulus intensity?). How is the amplitude related to the maximal response? PPR is strongly dependent on the stimulus strength; the effect of baclofen might be also dependent on the stimulation intensity. Could you comment this?

This was an inappropriate choice of wording, as it should have been “PSC amplitude” and not “intensity”. This has been corrected.

Regarding the relationship between PSC amplitude and Baclofen response or PPR, we did not observe a statistically significant effect of EPSC or IPSC amplitude on either property. Please see the plots in Author response image 2.

**Author response image 2. respfig2:** Dependence of baclofen mediated presynaptic inhibition and paired-pulse ratio on the EPSC and IPSC ampli-tudes. (**A**) Baclofen-induced presynaptic inhibition of EPSCs (filled circles) and IPSCs (open circles) plotted as a function of PSC amplitude. Based on a linear regression analysis, neither EPSCs (F(d.f. 1,13) = 2.5, p = 0.14, F-test, solid line) nor IPSC s(F(d.f. 1,7) = 0.00002, p = 0.99, F-test, dashed line) displayed a dependence of the bac-lofen response on the PSC amplitudes. (**B**) EPSC and IPSC amplitude effect on the degree of paired pulse ratio observed. Based on a linear regression analysis, neither EPSCs (F(d.f. 1,13) = 2.1, p = 0.17, F-test) nor IPSC s(F(d.f. 1,7) = 2.0, p = 0.20, F-test) displayed a strong interaction of PPR and synaptic strength.

– Subsection “Paired recordings from synaptically coupled CA1 PC-IN pairs”: Could you explain why you use two different concentration of EGTA: 10 and then 0.5 mM?

A lower concentration of EGTA was used for presynaptic experiments to prevent excessive Ca^2+^ buffering saturating endogenous buffering mechanisms. A clarification has been added to the Materials and methods.

7) Discussion:Several sentences are unclear. Please revise for lucidity: Paragraph one and two: What is "them" related to? SOM-Ins or PCs or both?

We have reworded these sentences and hope that we have addressed the reviewers concerns well in the revised version.

– Figure 5: E: it is unclear how many cells are there in fact. If points are overlaid, please arrange them in the way you use in Figure 4.

We would prefer to keep the 5E layout, consistent with the panels C and D, but added the numbers of cells to the legend.

– Figure 7: Consider different, more distinct colors, clarify legend.

We have re-coloured Figure 7 according to the Colour Universal Design scheme for 3 channel colours. We have also revised the legend.